# Uncovering the Mechanisms: The Role of Biotrophic Fungi in Activating or Suppressing Plant Defense Responses

**DOI:** 10.3390/jof10090635

**Published:** 2024-09-05

**Authors:** Michel Leiva-Mora, Yanelis Capdesuñer, Ariel Villalobos-Olivera, Roberto Moya-Jiménez, Luis Rodrigo Saa, Marcos Edel Martínez-Montero

**Affiliations:** 1Laboratorio de Biotecnología, Facultad de Ciencias Agropecuarias, Universidad Técnica de Ambato (UTA-DIDE), Cantón Cevallos Vía a Quero, Sector El Tambo-La Universidad, Cevallos 1801334, Ecuador; 2Natural Products Department, Centro de Bioplantas, Universidad de Ciego de Ávila Máximo Gómez Báez, Ciego de Ávila 65200, Cuba; yaneliscr@gmail.com; 3Facultad de Ciencias Agropecuarias, Universidad de Ciego de Ávila Máximo Gómez Báez, Ciego de Ávila 65200, Cuba; villalobos.olivera@gmail.com; 4Facultad de Diseño y Arquitectura, Universidad Técnica de Ambato (UTA-DIDE), Huachi 180207, Ecuador; rc.moya@uta.edu.ec; 5Departamento de Ciencias Biológicas y Agropecuarias, Facultad de Ciencias Exactas y Naturales, Universidad Técnica Particular de Loja (UTPL), San Cayetano Alto, Calle París s/n, Loja 1101608, Ecuador; lrsaa@utpl.edu.ec

**Keywords:** biotrophic fungi, effectors, ethylene, jasmonic acid, plant defense responses, salicylic acid

## Abstract

This paper discusses the mechanisms by which fungi manipulate plant physiology and suppress plant defense responses by producing effectors that can target various host proteins. Effector-triggered immunity and effector-triggered susceptibility are pivotal elements in the complex molecular dialogue underlying plant–pathogen interactions. Pathogen-produced effector molecules possess the ability to mimic pathogen-associated molecular patterns or hinder the binding of pattern recognition receptors. Effectors can directly target nucleotide-binding domain, leucine-rich repeat receptors, or manipulate downstream signaling components to suppress plant defense. Interactions between these effectors and receptor-like kinases in host plants are critical in this process. Biotrophic fungi adeptly exploit the signaling networks of key plant hormones, including salicylic acid, jasmonic acid, abscisic acid, and ethylene, to establish a compatible interaction with their plant hosts. Overall, the paper highlights the importance of understanding the complex interplay between plant defense mechanisms and fungal effectors to develop effective strategies for plant disease management.

## 1. Introduction

Biotrophic fungi (BF), such as rusts, powdery mildews, and smuts, are specialized pathogens that infect living plant cells without immediate cell death, significantly impacting agriculture. These fungi belong to different lineages: rusts (Pucciniales) [1] and smuts (Ustilaginales) [2] within the Basidiomycota, and powdery mildews (Erysiphales) within the Ascomycota [3].

Rust fungi:

Rust fungi, among the most diverse fungal orders, include about 8000 species affecting a wide range of host plants, from ferns to monocots and gymnosperms to angiosperms [4]. This broad host range underscores their adaptation to parasitizing living plant tissues, making them significant threats to crops and forestry globally [5]. Efforts to develop resistant cultivars are ongoing, though fungi like *Puccinia graminis* f. sp. *tritici* have evolved ways to overcome these defenses [6]. Research focuses on understanding how these fungi manipulate host immunity, which is supported by discoveries of numerous effector proteins [7].

Powdery mildew fungi:

Powdery mildew fungi from the Erysiphales order within the Ascomycota are obligate biotrophic pathogens infecting aerial plant tissues [8]. With around 9838 species across 1617 genera, such as *Blumeria graminis*, these fungi cause notable yield losses [9]. They target epidermal cells, requiring attachment and penetration through cuticle and cell walls to establish biotrophic interactions [10]. The development of appressoria is essential for breaching host defenses [11]. Despite challenges in cultivation and genetic manipulation, genomic and transcriptomic studies are uncovering the molecular mechanisms behind powdery mildew pathogenesis [12].

Smut fungi:

Smut fungi, including *Ustilago maydis*, are plant pathogens that primarily affect monocotyledonous species, including important cereal crops [13]. They produce darkly pigmented teliospores within floral structures, impacting plant reproduction [14]. Exhibiting dimorphism with yeast-like and filamentous phases, smut fungi colonize vascular tissues and cause symptoms only when systemic infection leads to localized tumors on plant parts [15]. Effector proteins secreted into host tissues play a crucial role in modulating defense responses, showcasing an evolutionary strategy to evade host immunity [16].

BF elicit plant defense responses, which are categorized into well-understood and less-understood pathways [17]. They use various strategies to colonize host plants while manipulating the host immune system [18]. Some BF also activate plant defenses [19]. Key mechanisms include effector-mediated interference [20], hormone signaling manipulation [21], inhibition of programmed cell death [22], suppression of pattern-triggered immunity, and modulation of transcriptional regulation [23].

This review aims to elucidate the mechanisms by which BF impact plant hosts, focusing on several key aspects. First, it explores how BF affect plant hosts through various strategies such as manipulating plant hormones, inhibiting programmed cell death, detoxifying reactive oxygen species (ROS), and modulating nitric oxide (NO) biosynthesis and signaling. Following this, the review delves into the role of effectors from BF in suppressing plant defense responses. In addition, it examines how these effectors may induce plant defense responses through effector-triggered immunity (ETI). The discussion then shifts to the role of plant hormones in modulating defenses against BF, addressing both suppression and activation mechanisms. Another critical aspect covered is how BF overcome ROS to evade plant defense responses. The review also provides insights into the co-evolutionary dynamics between BF and plants, highlighting the evolutionary arms race that shapes their interactions and adaptive strategies. Conclusively, it integrates these findings into a broader context, unraveling the complexity of plant–fungal interactions and emerging mechanisms, and offers potential strategies for plant disease management.

## 2. Mechanisms of BF Impact on Plant Hosts

### 2.1. Developing Specialized Structures

Biotrophic fungi (BF) use an array of intricate mechanisms to invade and sustain their presence in plant hosts [19]. A primary strategy involves the development of specialized structures, such as appressoria and haustoria, which enable the fungi to penetrate plant tissues and extract nutrients from the host cells [24,25]. Additionally, BF secrete effector proteins that manipulate host cellular processes, suppress immune responses, and interfere with hormone signaling pathways. For example, effectors may specifically target components of salicylic acid (SA) [26] and jasmonic acid (JA) pathways [27]. Furthermore, BF modulate the plant’s oxidative stress environment by influencing the activity of antioxidant enzymes, helping to suppress defense responses and creating a more favorable environment for colonization [28].

In rust fungi, urediniospores or teliospores develop germ tubes that sense physical and chemical cues from the host plant’s surface [29]. The germ tube tip swells and develops into a specialized structure called an appressorium that is typically flattened and melanized (darkly pigmented) and adheres to the host plant’s surface [30].

Within the appressorium, a specialized infection structure called a penetration peg (narrow, hypha-like structure that extends from the appressorium and penetrates the host plant’s cell wall) [31] by a combination of mechanical (high internal turgor pressure) [32] and enzymatic activity (cutinase [33], cellulase [34], pectinase [35], xylanase [36] and protease [37] to breach the host plant’s cell wall, facilitating the invasion of the host tissue.

Once the penetration peg has successfully entered the host plant’s cell, it develops a specialized infection structure called a haustorium that allows it to obtain nutrients from the host plant’s cells [38] and secrete effector proteins to the host cytoplasm, enabling the fungus to establish a successful infection [39].

Powdery mildew fungus appressoria are generally more flattened and are heavily pigmented, often appearing dark brown or melanized with an adhesive ring to adhere tightly to the host plant’s surface [40]. Penetration peg forms within the appressorium and then breaches the cell wall, allowing the fungus to enter the host plant’s cells [41]. During the infection of *Blumeria graminis*, enzymes like glycoside hydrolase and degradation-associated carbohydrate-active enzymes were recorded [42].

When teliospores germinate for a promycelium and undergo mitotic divisions, smut fungi eventually bud off haploid cells known as sporidia [43]. When two compatibles haploid sporidia detect each other through pheromone signaling, they develop conjugation tubes directed toward one another [44]. Smut fungi then sense plant surface cues and induce specialized infection structures called appressoria, with transmembrane receptors Sho1 and Msb2 playing essential roles in perceiving external stimuli [45]. To breach the plant cell wall, smut fungi secrete various plant cell wall-degrading enzymes like endo-β-1,4-glucanases, exo-β-1,4-glucanases [46], xylanases [47], mannanases [48], β-glucosidases [49], polygalacturonases [50], pectin lyases [51], and pectate lyases [52].

### 2.2. Secrete Effector Proteins in Plants

Effector proteins that can be directed to the host chloroplast by mimicking the plant’s own sorting signals are like suppressors of the host’s RNA interference (RNAi) machinery, targeting conserved cellular proteins that are essential components of the plant’s defense pathways [53]. By disabling the host’s RNAi system, the fungal effectors can effectively disrupt the plant’s ability to organize a robust immune response, ultimately facilitating the pathogen’s successful colonization and proliferation within the host [54].

Biotrophic fungal pathogens have evolved sophisticated effector proteins that can directly bind to the promoters of host defense genes, thereby modulating their transcriptional processes and leading to the suppression of the plant’s immune responses [55]. Furthermore, fungal effectors can also camouflage themselves as host modulators, diverting the metabolic flux of various compounds within the plant, resulting in a deficiency of crucial precursors or defense-related compounds [56].

BF induce the host to produce anti-cell death factors or hijack the plant’s own cell death regulatory machinery to prevent the activation of programmed cell death [57], which would otherwise limit the pathogen’s ability to establish a successful infection [58]. By effectively suppressing cell death signaling [59], biotrophic fungi can create a hospitable environment within the host [60], allowing them to thrive and proliferate without triggering the plant’s defensive cell death responses [61].

By suppressing ROS accumulation, biotrophic fungi can create a more favorable environment for their growth and proliferation within the host plant [21]. Specific fungal effectors, such as TalSP [62] and the nudix hydrolases TaNUDX23 [63], have been identified as playing crucial roles in this process.

Biotrophic fungal effectors can target and disrupt the host’s pattern recognition receptors (PRRs) that are responsible for detecting pathogen-associated molecular patterns (PAMPs) [64]. By interfering with the activation or signaling of these PRRs, the effectors can prevent the initiation of the PTI cascade, effectively shutting down the plant’s first line of defense [53].

Additionally, some fungal effectors can directly inhibit the callose deposition process. Callose is a polysaccharide that is rapidly deposited at the site of attempted pathogen invasion, forming a physical barrier to restrict pathogen entry [7]. Certain effectors may interfere with the enzymatic machinery responsible for callose synthesis or may trigger the host’s own negative regulators of callose deposition [65].

Some effectors may promote the degradation of the host’s Adenosine kinase (ADK) proteins either by direct targeting or by inducing host processes that degrade the enzymes [65]. The reduced levels of ADKs lead to a decrease in salicylic acid production and the subsequent suppression of PTI [66].

Fungal effectors can target and disrupt the host’s machinery, which is responsible for the formation of defense-related vesicles, such as the endoplasmic reticulum and Golgi apparatus [67]. By interfering with the proper assembly and cargo loading of these vesicles [68], the effectors can prevent the transport and secretion of defense proteins [69], enzymes [70], and antimicrobial compounds [53].

BF effectors promote enhanced plasmodesmatal flux, facilitating the intercellular movement of nutrients and suppressing host defense responses [71]. Modification in the structure or regulation of plasmodesmata, modified cytoplasmic channels that connect plant cells [72,73]. By increasing the permeability and size exclusion limit of plasmodesmata, the fungal effectors can enable the passage of nutrients and signaling molecules while limiting the transport of defense-related compounds [74].

Biotrophic fungi have evolved effector proteins that can directly interact with and manipulate host transcription factors to suppress plant defense responses [24]. These effectors may bind to and sequester key transcription factors that are responsible for activating defense-related genes, preventing them from initiating the transcriptional programs necessary for the plant’s immune response [24]. Alternatively, the fungal effectors may act as transcriptional co-regulators, interfering with the ability of defense-related transcription factors to bind to and activate their target genes [74].

Fungal effectors may directly bind to and mask the Avr proteins, preventing their detection by the corresponding host resistance (R) proteins [75]. Additionally, the effectors may interfere with the signaling pathways that normally lead to the activation of ETI upon Avr protein recognition, disrupting the downstream defense responses [76].

*Blumeria graminis* f. sp. hordei accumulates 3-hydroxykynurenine, a redox-active substance, which facilitates the cross-linking of the pathogen to the host surface [21]. Bgh also expresses a secreted catalase enzyme that is essential for removing hydrogen peroxide produced by the host plant [77]. The removal of hydrogen peroxide by catalase prevents the host from cross-linking its cell wall as a defense mechanism against pathogen penetration [78].

Some rust fungi have evolved the ability to produce their own SOD enzymes. This allows them to neutralize the ROS produced by the plant, reducing the oxidative stress on the fungus [79]. By scavenging superoxide radicals, the fungal SOD can help the pathogen evade or suppress the plant’s initial defense response, allowing the infection to become established [80].

Glycine-serine-rich effector, PstGSRE4, produced by the wheat stripe rust fungus *Puccinia striiformis* f. sp. *tritici* (Pst), has been shown to inhibit the enzymatic activity of the wheat copper-zinc superoxide dismutase (TaCZSOD2) [81]. This wheat superoxide dismutase enzyme acts as a positive regulator, enhancing the plant’s resistance against the Pst pathogen [79]. By targeting and suppressing the activity of TaCZSOD2, PstGSRE4 effectively disrupts the host’s ability to mount an effective oxidative defense response [79]. This strategy enables the rust fungus to evade or dampen the plant’s initial defense mechanisms, facilitating the establishment and progression of the infection.

*Ustilago maydis* has evolved a highly effective virulence mechanism mediated by its secreted effector protein, Pep1 [82]. Studies have demonstrated that Pep1 plays a crucial role in suppressing the host’s oxidative burst response during the early stages of infection [83]. Mechanistically, Pep1 functions by directly inhibiting the activity of apoplastic peroxidases in the plant’s extracellular space. Peroxidases are key enzymes involved in the generation of reactive oxygen species (ROS), which are typically produced by the host as a defense response against invading pathogens [7]. By neutralizing the activity of these apoplastic peroxidases, the Pep1 effector effectively dampens the plant’s ability to mount an oxidative burst, a critical component of the innate immune system [84]. Some powdery mildew species have been found to secrete their own peroxidase enzymes, which can break down the H_2_O_2_ and other ROS produced by the plant [77]. By neutralizing the ROS, the fungal peroxidases can help the pathogen evade or suppress the plant’s initial oxidative defense response, allowing the infection to establish and progress [78].

*Puccinia striiformis* f. sp. *tritici* (Pst) secreted catalase enzyme that plays a crucial role in the removal of hydrogen peroxide (H_2_O_2_) produced by the plant as part of its oxidative defense. By effectively neutralizing the H_2_O_2_, the fungal catalase enables growth and spread during host infection [77].

Through these actions, effector molecules help pathogens evade host defenses and establish a compatible interaction for successful infection. Table 1 lists notable examples of effector molecules produced by various BF along with their references.

### 2.3. Activation or Deactivating Antioxidant Enzymes

BF manipulate plant hormones and antioxidant enzymes to evade plant defenses and establish infection. A key strategy involves secreting effector molecules that mimic or interfere with SA and jasmonic acid (JA) signaling pathways, disrupting their balance to dampen defense responses and create a favorable environment for fungal growth [118].

BF also counteract reactive oxygen species (ROS), such as hydrogen peroxide (H_2_O_2_) and superoxide radicals (O_2_˙^−^), produced by plants as part of their defense mechanisms [119]. To neutralize ROS, BF produce antioxidant enzymes like catalase (CAT) and peroxidase (POX) and secrete effector molecules that scavenge ROS or inhibit ROS-generating machinery [120]. This detoxification process helps BF evade plant defenses and maintain compatibility with the host [121]. Additionally, BF interfere with nitric oxide (NO) biosynthesis and signaling pathways, reducing NO activity through inhibition and modulation of NO biosynthesis and production of NO-scavenging enzymes [122].

Certain powdery mildew species, such as *Erysiphe necator* and *Golovinomyces orontii*, secrete POX enzymes to break down H_2_O_2_ and other ROS produced by the plant. This neutralization helps the pathogen evade the plant’s initial oxidative defense response, allowing infection to establish and progress [123].

*Blumeria graminis* f. sp. *hordei* (Bgh) accumulates 3-hydroxykynurenine, a redox-active substance that facilitates cross-linking to the host surface [100]. Bgh also expresses a secreted CAT enzyme to remove H_2_O_2_ produced by the host, preventing the host from cross-linking its cell wall as a defense mechanism against pathogen penetration [124].

Rust fungi have evolved the ability to produce superoxide dismutase (SOD) enzymes, neutralizing ROS and reducing oxidative stress [125]. The wheat stripe rust fungus, *Puccinia striiformis* f. sp. *tritici* (Pst), produces an effector named PstGSRE4, which inhibits the activity of the wheat copper-zinc SOD (TaCZSOD2), a positive regulator of plant resistance. By targeting TaCZSOD2, PstGSRE4 disrupts the host’s oxidative defense, facilitating infection [79].

Smut fungi, such as *Ustilago maydis*, utilize a secreted effector protein, Pep1, to suppress the host’s oxidative burst response during early infection stages. Pep1 inhibits the activity of apoplastic POXs in the plant’s extracellular space, dampening the plant’s ability to mount an oxidative burst, a critical component of its innate immune response [126].

## 3. Mechanism by Which Effectors May Suppress Plant Defense Responses

Effector molecules, secreted by BF, play a crucial role in manipulating plant physiology and suppressing defense responses. While some effectors can trigger immune responses, others suppress these defenses to facilitate pathogen colonization. Effectors can activate ETI, a specific branch of the plant immune system that recognizes pathogen effectors and often leads to strong, durable immunity [127,128].

BF employ effector-mediated interference to disrupt host plant defense signaling pathways [20]. This interference targets downstream signaling components involved in defense responses. Effectors can disrupt gene activation or interfere with signaling cascades, leading to defense protein expression, including blocking transcription factor activation or inhibiting enzymes involved in defense signaling [56,129,130].

Effector molecules can specifically target and interact with various host proteins, including RLKs [131] and transcription factors [132,133,134], to promote fungal growth while suppressing plant defense responses [27]. RLKs, which perceive and transmit signals, can be directly interacted with by BF effectors, disrupting their signaling domains [135].

Table 2 provides a comprehensive summary of various genes and proteins involved in plant defense responses against a range of pathogens. It highlights different types of proteins such as RLPs (Receptor-Like Proteins), LRR-RLKs (Leucine-Rich Repeat Receptor-Like Kinases), NLRs (Nucleotide-Binding Leucine-Rich Repeat Receptors), SERKs (Somatic Embryogenesis Receptor Kinases), WRKY transcription factors, and syntaxins. Each entry details the specific plant species in which these proteins are found, the corresponding pathogen they defend against, and their role in the plant’s immune response. The table underscores the complex interplay between plant immune receptors and pathogen effectors, illustrating how plants recognize and respond to pathogenic threats to trigger effective defense mechanisms. This detailed information serves as a valuable resource for understanding plant–pathogen interactions and the molecular basis of plant immunity.

### Specific Examples and Interactions

Various mechanisms by which BF effectors interact with plant defense systems are exemplified by the following specific interactions involving different receptor proteins:

LRR-RLKs: In wheat, Lr10 and Lr21 recognize specific avirulence effectors and trigger defense responses [155].

NLRs: Proteins like Sr33 and Sr35 in wheat recognize specific rust fungus effectors, triggering defense responses [156]. The wheat resistance genes Sr33 and Sr35 confer race-specific resistance against *Puccinia graminis* f. sp. *tritici* where a specific resistance (R) gene, such as Sr33 or Sr35, recognizes a corresponding avirulence (Avr) effector molecule produced by a specific race or strain of the pathogen [157]. This recognition triggers a robust defense response that effectively blocks the development and spread of the recognized pathogen race within the host plant. In the case of Sr33 (2) and Sr35 [157], these R genes have been shown to specifically recognize and respond to certain Avr effectors produced by various *Puccinia graminis* f. sp. *tritici* races [158]. The resistance conferred by these genes is effective against some, but not all, stem rust pathogen races, as the pathogen can evolve to evade recognition by these R genes through mutations in the corresponding Avr genes [159].

Effector molecules can also interact with transcription factors and key gene expression regulators to suppress defense genes. For instance, Stb6 in wheat is targeted by the effector AvrStb6 from the wheat stripe rust pathogen [160], and Sge1 in maize is targeted by the smut fungus *U. maydis* effector Pit2, preventing the activation of defense-related genes [161].

Table 3 provides a comprehensive overview of effector-mediated suppression of plant defense responses. This structured summary highlights the key mechanisms and specific examples of how effector molecules interfere with plant immunity. The table illustrates various strategies pathogens employ, including interacting effectors with specific host proteins, to overcome plant defense systems. Presenting this information in a concise, tabular format provides a clear and accessible reference for understanding the complex interplay between pathogen effectors and plant immune responses.

Figure 1 illustrates the strategies used by various BF in penetrating, colonizing, and multiplying within plant cells and tissues. The figure provides detailed examples of the mechanisms employed by rust fungi, powdery mildew fungi, and smut fungi, including the roles of specific effector proteins. Rust fungi produce appressoria at the tip of infection pegs, using enzymes and mechanical force to breach the plant cuticle and cell wall. Effector proteins such as AvrSr35 and PgtSR1 facilitate evasion and suppression of plant immune responses. Powdery mildew when conidia land on the plant surface and germinate, producing a germ tube that differentiates into an appressorium. Effector proteins like AVRA1 and BEC1016 silence plant defense responses. Smut fungi using teliospores germinate to produce infection hyphae that develop appressoria. Effector proteins such as Pit2 and Pep1 promote systemic colonization and nutrient extraction.

Table 4 presents the strategies employed by different BF, detailing their penetration mechanisms, key effectors, nutrient acquisition mechanisms, and reproduction strategies. This table enhances the understanding of how various fungi interact with host plants and adapt their infection strategies. Presenting this information in a structured and detailed format provides a clear and accessible reference for researchers studying plant–pathogen interactions. The comprehensive overview of effector-mediated suppression mechanisms and the specific examples of fungal strategies will aid in the development of new approaches to enhance plant resistance to fungal pathogens.

## 4. Mechanism by Which Effectors Induce Plant Defense Responses ETI

ETI refers to the plant’s ability to recognize and respond to pathogenic microorganisms through the detection of specific molecules called effectors, which are secreted by the pathogens [166,167]. ETI involves the recognition of these effectors by plant resistance proteins (R proteins), leading to the activation of a signaling cascade [132]. This cascade induces defense responses such as the hypersensitive response (HR), which involves localized programmed cell death at the site of infection and the production of reactive oxygen species (ROS) and antimicrobial compounds [168]. This sophisticated defense mechanism allows plants to combat pathogens effectively [169].

### 4.1. Example: Puccinia graminis in Wheat

*P. graminis*, the causal agent of stem rust disease in wheat, exemplifies how ETI functions. Upon infecting wheat, *P. graminis* releases specific effectors into plant cells to manipulate their physiology, facilitating infection [170]. However, plants have evolved a two-tiered immune system to counteract these pathogenic effects: Innate Immunity provides a basal level of defense against a broad range of pathogens [162]. ETI: Activated upon recognition of specific effectors by plant resistance proteins (R proteins) [171].

### 4.2. Gene-for-Gene Interaction

The recognition of pathogen effectors by R proteins follows a gene-for-gene interaction model, where each R protein recognizes a corresponding effector molecule produced by the pathogen [172]. In the *P. graminis*-wheat pathosystem, specific R proteins in wheat recognize specific effectors secreted by the fungus [173].

R Genes in Wheat:

Sr33 and Sr35 provide resistance against multiple races of *P. graminis*. Sr33 is known for its broad-spectrum resistance, while Sr35, identified in the wheat variety “Thatcher”, is highly effective against a wide range of stem rust races. Both genes are extensively utilized in wheat breeding programs for their effectiveness and durability [174]. Sr39 provides resistance against various races of stem rust [175]. Sr21 offers effective resistance against specific races of *P. graminis* but may be susceptible to certain strains [176].

### 4.3. ETI against Other Pathogens

*Puccinia striiformis*: The causal agent of stripe rust disease involves ETI through the recognition of specific effectors by R proteins in the host plant [90]. Several R genes, such as Yr5, Yr10, Yr15, and Yr17, confer resistance against *P. striiformis.* Yr5 confers resistance through ETI [177]. Yr10 is another R gene that confers resistance against *P. striiformis* [178]. Additionally, Yr15 and Yr17 provide resistance against a wide range of *P. striiformis* races [179,180].

*Blumeria graminis*: The family of mildew resistance locus a and o (Mla and Mlo) genes in barley plays a crucial role in recognizing specific effectors produced by *B. graminis*, triggering immune responses that lead to resistance against powdery mildew [100,181]. Similarly, Pm3 (Powdery mildew resistance 3) genes in wheat are involved in ETI-mediated resistance [182].

Barley, like wheat, is susceptible to infection by the fungal pathogen *Puccinia graminis* f. sp. hordei, the causal agent of barley stem rust [183]. Plants have evolved a variety of resistance (R) genes that can recognize and respond to specific pathogen effectors or molecules, triggering a defense response that prevents or limits disease development [184].

The resistance conferred by these R genes is often race-specific, meaning that the R gene can only recognize and respond to certain races or strains of the pathogen that carry the corresponding avirulence (Avr) gene or molecule [185]. In the case of barley stem rust, specific R genes have been identified that can provide resistance against certain races of *P. graminis* f. sp. *hordei*, but not others [186]. This is due to the evolutionary arms race between the plant and the pathogen, where the pathogen constantly evolves new virulence factors (Avr genes) to evade recognition by the host’s R genes, while the plant acquires new R genes to maintain resistance [187].

For example, the barley resistance gene Rpg1 confers resistance against some, but not all, races of *P. graminis* f. sp. *hordei* [188]. The Rpg1 protein recognizes and responds to specific Avr effectors produced by certain pathogen races, triggering a defense response that effectively blocks the development of the recognized races. However, as the pathogen evolves and acquires mutations in the Avr genes, it can become virulent against the Rpg1-mediated resistance, rendering the host plant susceptible to particular pathogen races [189].

*Ustilago maydis*: ETI against *U. maydis* involves the recognition of specific effectors by plant R genes, leading to robust immune responses. Examples include Rp1-D, Rp3, and Rp6 in maize by recognizing specific effectors such as Pep1 and Cmu1 and activating defense responses [190].

### 4.4. Comprehensive Overview and Visual Aids

To provide a comprehensive overview, Table 5 details the mechanisms and examples of ETI and gene-for-gene interactions in plant–pathogen interactions. This table presents a valuable reference for researchers interested in plant immunity mechanisms and their practical applications. This understanding is crucial for elucidating the complex interactions between plants and pathogens and may offer insights into developing strategies for disease resistance in crops.

Table 6 highlights various BF and their corresponding plant defense mechanisms, specifically focusing on the types of R genes involved and referencing key studies that have identified these interactions. This detailed information is valuable for researchers focusing on BF and can be directly applied to breeding programs and genetic engineering aimed at improving resistance to specific BF.

To further illustrate the effector proteins secreted by BF that suppress plant defense responses, Figure 2 provides insights into the diverse array of effector molecules deployed by these fungi to undermine host defenses.

Furthermore, to illustrate the intricate mechanisms underlying effector-induced plant defense responses, particularly ETI, Figure 3 provides a visual representation of plant receptor proteins and their pivotal role in detecting BF and eliciting plant defense responses. This figure elucidates the intricate interplay between plant receptors and pathogenic effectors, shedding light on the molecular mechanisms driving ETI.

## 5. Plant Hormones Suppress Plant Defense Responses against BF

### 5.1. Jasmonic Acid (JA)

BF employ mechanisms that manipulate the JA pathway, a critical signaling pathway in plant defense against various pathogens [204]. By secreting effector molecules, these fungi interfere with JA signaling, suppressing the activation of plant defense responses and thereby ensuring their successful colonization of host plants [205]. This intricate manipulation of the JA pathway facilitates a compatible interaction between BF and their plant hosts, underscoring the complex interplay between pathogens and plant defense mechanisms [206].

Plant hormones, including JA, play complex roles in modulating plant defense responses against pathogens such as *Puccinia graminis*, the causative agent of stem rust in wheat [207]. While some hormones enhance plant defense, others may suppress or attenuate defense mechanisms [208].

*Puccinia striiformis* suppresses plant defense responses by manipulating the JA pathway [209]. The fungus secretes effector molecules that interfere with JA signaling, leading to the downregulation of JA-responsive genes involved in plant defense [210]. This manipulation weakens the host plant’s defense, allowing *P. striiformis* to establish and spread within the plant [90].

Similarly, *Melampsora lini* and *Phakopsora pachyrhizi* secrete effector molecules that interfere with JA signaling, resulting in the downregulation of JA-responsive genes and facilitating successful fungal colonization and disease establishment [211].

BF, such as *Puccinia*, may secrete effector proteins that directly or indirectly manipulate the host plant’s defense signaling pathways [212]. These effectors interact with components of the JA signaling pathway, altering its activation or suppressing downstream defense responses [213]. By interfering with JA signaling, the pathogen dampens the plant’s defense mechanisms, enabling the establishment and maintenance of a compatible interaction [214].

JA is integral in regulating plant defense, particularly against insect herbivores and necrotrophic pathogens [215]. However, *P. graminis* can manipulate JA signaling to suppress plant defenses by producing effector molecules that interfere with JA pathways [216].

BF may produce effectors that promote the synthesis or accumulation of JA while suppressing other defense-related hormones, such as SA [217,218]. This hormonal crosstalk shifts the plant’s defense response away from SA-mediated defenses, which are typically effective against biotrophic pathogens, toward a JA-dominated response [219]. By promoting JA and inhibiting SA, the pathogen undermines the plant’s defense against biotrophic fungi [219].

Traditionally, powdery mildew fungi have been associated with suppressing the SA pathway; recent research indicates that *Blumeria graminis* may also interfere with the JA pathway to manipulate plant defense responses [220]. Although the exact mechanisms are not fully understood, studies have shown that the fungus secretes effector molecules that modulate JA signaling and suppress JA-responsive defense genes [53].

Negative regulators such as JASMONATE ZIM-DOMAIN (JAZ) proteins typically repress JA-responsive genes and prevent the activation of defense responses [215]. By suppressing the activity of these inhibitors, the pathogen enhances JA signaling and dampens the plant’s defense mechanisms [221].

*Ustilago maydis* secretes effector molecules that target and interact with JAZ proteins, leading to their degradation or sequestration [222]. JAZ proteins normally act as repressors of JA-responsive genes in the absence of JA signaling [223]. By manipulating JAZ proteins, BF disrupt the repression of JA-responsive genes, effectively suppressing plant defense responses reliant on JA-mediated pathways [224]. This manipulation facilitates a compatible interaction with the host plant, aiding successful colonization and disease development [21].

BF often induce changes in host plant metabolism to create a nutrient-rich environment favorable for their growth and development [225]. These metabolic alterations can indirectly affect JA signaling by diverting resources or interfering with the synthesis and perception of defense-related compounds [226]. By manipulating host metabolism, the pathogen influences the activation and effectiveness of JA-mediated defense responses [227].

### 5.2. Salicylic Acid (SA)

BF produce effector proteins that directly target components of the SA pathway [228]. These effectors interfere with SA synthesis, perception, or SA-dependent signal transduction, thereby disrupting SA-mediated defense responses [229]. For instance, some effectors inhibit the enzymatic activity of key enzymes involved in SA biosynthesis or modify host proteins within the SA signaling pathway [230]. 

BF secrete effectors that suppress SA production in infected plants by targeting and inhibiting key enzymes involved in SA biosynthesis. This reduction in SA levels weakens the plant’s defense response, facilitating fungal infection [113].

Biotrophic pathogens manipulate the balance between different phytohormones, including SA, JA, and ethylene (ETH) [210]. Pathogens may produce effectors that promote the synthesis or accumulation of JA or ETH while simultaneously suppressing SA [231]. This hormonal crosstalk shifts the plant’s defense response away from SA-mediated defenses, favoring alternative pathways that are less effective against BF [232].

BF produce effectors, such as Pit2 and Cmu1, that interfere with SA signaling pathways in host plants [232,233]. These effectors target and disrupt the activity of key components involved in SA perception or downstream signaling, such as SA receptors or transcription factors [26]. By inhibiting SA signaling, the pathogen prevents the activation of defense-related genes, thereby dampening the plant’s defense response against biotrophic infection [234].

*Puccinia* species produce effector proteins that directly target components of the SA pathway in host plants [53]. These effectors interfere with SA biosynthesis, perception, or signaling by inhibiting enzymes involved in SA biosynthesis or modifying host proteins within the SA signaling pathway, thus disrupting SA-mediated defense responses [235,236].

Biotrophic pathogens like *Puccinia* manipulate the balance between defense-related hormones, such as SA and JA [237]. They produce effectors that promote JA synthesis or accumulation while suppressing SA. JA is typically associated with defense against necrotrophic pathogens and herbivores, whereas SA is more closely linked to defense against BF [229,238]. By favoring JA over SA, *Puccinia* redirects the plant’s defense response away from effective SA-mediated defenses, weakening the host’s ability to combat biotrophic infection [238].

*Puccinia* effectors target and disrupt components involved in SA perception or downstream signaling, such as SA receptors or transcription factors. By inhibiting SA signaling, the pathogen hampers the activation of defense-related genes and impairs the plant’s ability to mount an effective defense response against biotrophic infection [81,92].

Powdery mildew fungi have evolved mechanisms to manipulate SA and suppress plant defense responses [239]. These pathogens produce effector proteins that interfere with the plant’s ability to produce SA or its precursors [240]. Additionally, BF might inhibit SA synthesis, thereby preventing the activation of plant defense responses [19].

BF produce proteins or molecules that interfere with SA-mediated signaling pathways. By disrupting these pathways, powdery mildew fungi inhibit the transmission of defense signals and suppress the plant’s immune response [17]. Some powdery mildew species possess enzymes that modify SA or its derivatives, rendering them inactive or less potent. These detoxification mechanisms enable the pathogen to evade SA’s effects and undermine plant defenses [241].

*Blumeria graminis*, the causal agent of powdery mildew disease, primarily suppresses the SA pathway (Cmu1) rather than disrupting it. This BF manipulates plant defense responses by inhibiting SA-mediated defenses [242]. *B. graminis* secretes effector molecules that interfere with the SA signaling pathway, leading to the downregulation of SA-responsive genes involved in plant defense against pathogens [228]. By suppressing the SA pathway, the fungus evades recognition by the host plant’s immune system and establishes a successful infection [243].

*Tilletia caries* secretes effector molecules that interfere with the SA signaling cascade by targeting components of the SA pathway, such as SA receptors or downstream signaling molecules, inhibiting their normal function [244,245].

*U. maydis* has evolved mechanisms to suppress SA production in infected plants, thereby inhibiting defense responses by producing effector proteins that manipulate hormone signaling pathways, including those involved in SA biosynthesis [83]. These effectors interfere with the expression or activity of key enzymes involved in SA production, reducing SA levels within infected plants [246].

*U. maydis* secretes enzymes, such as salicylate hydroxylase and molecules like ZmCm2 and Cmu1, that directly inhibit enzymes involved in the SA biosynthesis pathway [247]. By disrupting SA synthesis, *U. maydis* impairs the plant’s defense response [248]. Additionally, the fungus can modify the infected plant’s metabolic pathways, redirecting precursors away from SA biosynthesis [249].

### 5.3. Abscisic Acid (ABA)

ABA is a plant hormone associated with diverse physiological processes, including stress responses [250]. In some cases, ABA has been implicated in the suppression of plant defense responses against *Puccinia graminis* [251].

Biotrophic pathogens can suppress ABA synthesis by producing effector proteins or molecules that inhibit ABA production in infected plant tissues [252]. By limiting ABA production, the pathogen disrupts normal plant stress responses and dampens defense signaling [253]. Additionally, these pathogens may manipulate ABA signaling pathways by producing effectors that interfere with ABA perception or transduction, thereby preventing the activation of ABA-mediated defense responses [252].

Some BF can redirect ABA signaling pathways to their advantage. They manipulate ABA-responsive genes or regulatory elements to create conditions favorable for their growth and colonization [254]. This manipulation can result in the suppression of defense-related genes or the activation of genes that benefit the pathogen [255].

Biotrophic pathogens often engage in complex crosstalk with multiple plant hormones, including ABA. They can modulate the balance between ABA and other hormones involved in defense responses, such as SA and JA. By shifting this hormonal balance, the pathogen can suppress SA- or JA-mediated defense pathways, which are antagonistic to ABA signaling [256]. For instance, in sugarcane plants inoculated with *P. kuehnii*, genes related to ABA metabolism were downregulated at 12 h after inoculation (hai) and repressed at 24 hai [257].

ABA-responsive genes, such as AAO3, are significantly induced in *ataf1* plants compared to wild-type plants following inoculation with *B. graminis* f. sp. *hordei* [258,259]. In *Hevea brasiliensis*, ABA can induce plant defense against *Erysiphe quercicola* and inhibit ABA biosynthesis by perturbing the localization of 9-cis-epoxycarotenoid dioxygenase 5 (HbNCED5), a key enzyme in ABA biosynthesis [53].

### 5.4. Ethylene (ETH)

BF have been found to disrupt the ETH signaling pathway to manipulate plant defense responses in their favor [260,261]. These fungi secrete effector molecules that interfere with various components of the ETH pathway, leading to the suppression of ETH-responsive genes involved in plant defense [262].

Several mechanisms have been proposed for this interference: (i) Production of effector proteins (e.g., Jsi1 from *Ustilago maydis*) that target and inhibit key enzymes involved in ETH biosynthesis, resulting in reduced ETH production [247]; (ii) Secretion of effector proteins that interfere with ETH receptor proteins or downstream signaling components, thereby inhibiting ETH signaling [263]; (iii) Disruption of the ETH pathway, leading to a diminished defense response [264].

By employing these mechanisms, BF can subvert the plant’s defense responses, as ETH plays a crucial role in activating defense mechanisms against pathogens [265]. This manipulation allows the fungi to establish compatible interactions with the host plant, suppress host immune responses, and ensure successful colonization and nutrient acquisition [266].

Studies have shown that *Blumeria graminis* secretes effector molecules that target and interfere with components of the ETH pathway, including ETH receptors and downstream signaling components [267]. This interference disrupts ETH signaling, leading to a dampened or altered ETH response in the host plant. By suppressing the ETH pathway, *B. graminis* effectively evades plant defenses. It produces effectors such as TuMYB46L and TuACO3, which regulate ETH biosynthesis in wheat, impacting defense responses associated with ETH signaling. This manipulation facilitates disease development and the establishment of a compatible interaction [268].

Effector proteins produced by *P. graminis*, the causative agent of stem rust disease, can also manipulate the ETH pathway to evade plant defense responses [269]. These effector proteins interact with components of the ETH pathway, such as ETH receptors and downstream signaling molecules, disrupting or modulating ETH signaling [270]. This manipulation potentially dampens or alters the plant’s defense responses, aiding in the pathogen’s successful infection.

Although *U. maydis*, the causal agent of corn smut disease, primarily interferes with the SA pathway rather than directly manipulating the ETH pathway, it still significantly impacts plant defense responses [271]. *U. maydis* secretes effector molecules that target and manipulate components of the SA pathway, suppressing SA-mediated defense responses in the host plant [26]. This interference underscores the complexity of hormonal crosstalk in plant–pathogen interactions, highlighting how pathogens can strategically manipulate multiple signaling pathways to their advantage.

Table 7 provides a structured overview of how BF manipulate various plant hormones, including JA, SA, ABA, and ETH, to suppress plant defense responses. It includes the mechanisms employed by the fungi, their effects on defense responses, and specific examples or details for each hormone-fungus interaction. This table highlights the intricate strategies BF employ to manipulate plant hormonal pathways, thereby facilitating their colonization and pathogenesis.

## 6. Overcoming ROS to Inactivate Plant Defense Responses

ROS are toxic molecules produced by plants as part of their defense response against pathogens. These molecules can damage pathogen cells, making ROS detoxification a critical strategy for successful colonization by BF [277].

*P. striiformis* have developed sophisticated mechanisms to detoxify ROS produced by their host plants [278]. During the interaction between plant and pathogen, the host’s defense response often includes the rapid production of ROS. To counteract this, *P. triticina* produces a suite of antioxidant enzymes, such as SOD, CAT, and POX, which convert ROS into less harmful compounds like water and oxygen [279].

In addition to enzymatic detoxification, *P. striiformis* secretes effector molecules that can interfere with the plant’s ROS production or signaling pathways. These effectors help the pathogen evade or suppress the host’s defense responses, further facilitating infection and colonization [280,281].

Similarly, *Phakopsora pachyrhizi* may also secrete effector molecules that can potentially manipulate host plant defenses, including the modulation of ROS production or signaling pathways [282]. These effectors have the potential to interfere with the plants’ ROS generation, which could help the pathogen evade or suppress the host’s defense responses [283].

By detoxifying ROS and evading the plants’ defense mechanisms, *Puccinia graminis* can enhance its ability to establish infection and colonize the host plant [284]. However, it is important to note that the precise mechanisms and effector proteins involved in ROS detoxification by *P. graminis* may vary among different strains or races of the pathogen, and further research is needed to fully understand these mechanisms [278].

*B. graminis*, the causal agent of powdery mildew disease, has evolved mechanisms to counteract ROS produced by the plant during defense responses [285]. The fungus produces antioxidant enzymes such as SOD, CAT, and POX that neutralize ROS by converting them into less harmful molecules [286]. These enzymes play a crucial role in detoxifying ROS and protecting the pathogen’s cells from oxidative damage [287].

*U. maydis* produces antioxidant enzymes such as SOD, CAT, and POX [288]. These enzymes play a crucial role in neutralizing ROS by converting them into less harmful compounds, thereby protecting the pathogen’s cells from oxidative damage. Glutathione is an essential antioxidant molecule found in many organisms. Studies have shown that *U. maydis* possesses a functional glutathione system, including the synthesis of glutathione and enzymes involved in glutathione-related processes [28]. Glutathione acts as a major antioxidant molecule, helping to mitigate the harmful effects of ROS [289].

*U. maydis* produces melanin pigments, which have been associated with antioxidant properties. Melanin can scavenge ROS and protect the pathogen’s cells from oxidative stress [290]. Melanin biosynthesis starts with the conversion of tyrosine, an amino acid, into a precursor molecule called L-DOPA (L-3,4-dihydroxyphenylalanine), which is further processed into melanin [291]. Melanin’s chemical structure allows it to capture and convert ROS into less harmful compounds, such as water, thereby preventing oxidative damage to the pathogen’s cellular components [292].

By scavenging ROS, melanin helps protect *U. maydis* hyphae from oxidative stress. Oxidative stress occurs when there is an imbalance between ROS production and the cellular antioxidant defense systems [293]. Melanin acts as an antioxidant in *U. maydis*, maintaining redox homeostasis and reducing the harmful effects of ROS on the pathogen’s cells [294].

Table 8 summarizes how BF overcome ROS, inactivating plant defense responses. This table provides a structured overview of how different BF, including *Phakopsora pachyrhizi*, *Puccinia graminis*, and *U. maydis*, overcome ROS, inactivating plant defense responses. It includes the mechanisms employed by each fungus, specific examples, and relevant references.

## 7. Co-Evolutionary Dynamic between BF and Plants

The co-evolutionary relationship between BF and plant defense mechanisms is pivotal for understanding host–pathogen interactions [19]. These ancient and intricate associations have evolved over millions of years, resulting in a dynamic “arms race” where pathogens develop strategies to overcome host defenses, driving the diversification of plant resistance traits [298]. Knowledge of this co-evolutionary dynamic is crucial for developing innovative approaches to sustainable crop protection [80]. By elucidating the specific evolutionary adaptations of plants and fungi, we can identify targets for genetic improvement of disease resistance and inform the design of novel fungicides and other control measures [299].

Understanding these relationships provides insights into the future trajectories of host–pathogen co-evolution, helping us prepare for and mitigate the emergence of new, virulent fungal pathogens [300]. A thorough understanding of the co-evolutionary arms race between BF and plant defense mechanisms is essential for safeguarding global food security and agricultural resilience in the face of evolving threats [121].

### 7.1. Plant Defense Mechanisms against BF

#### 7.1.1. Cell Wall Fortification (e.g., Callose Deposition)

One of the primary barriers against fungal invasion is the plant cell wall, fortified through the deposition of specialized compounds like callose [301]. Rapid and localized callose accumulation at pathogen entry sites effectively blocks fungal hyphae from penetrating the host tissue [302]. The dynamic remodeling of the cell wall in response to biotrophic fungal attacks is orchestrated by complex signaling cascades that rapidly detect and respond to the invading pathogen [303]. This multifaceted defense strategy, honed through eons of co-evolution, represents a critical adaptive trait that has allowed plants to thwart the advances of their fungal counterparts repeatedly [304].

#### 7.1.2. Production of Antimicrobial Compounds (Phytoalexins)

Plants have evolved a diverse array of defense mechanisms, including the production of specialized antimicrobial compounds known as phytoalexins [305]. These secondary metabolites are rapidly synthesized and accumulated at pathogen invasion sites, serving as a potent chemical barrier against fungal colonization [249]. The biosynthesis of phytoalexins is typically induced by the detection of pathogen-associated molecular patterns or the activation of defense signaling pathways, triggering a rapid and localized response to contain the fungal threat [306].

Over their co-evolutionary history, plants have diversified their phytoalexin repertoire, evolving an impressive chemical arsenal tailored to target the unique vulnerabilities of BF [307]. These compounds may disrupt fungal cell membranes, interfere with essential metabolic processes, or inhibit fungal virulence factors [308,309]. The structural diversity and targeted antimicrobial activities of phytoalexins reflect the ongoing evolutionary arms race between plants and fungal pathogens, where plants continuously adapt to stay ahead of their fungal counterparts [298].

The production of phytoalexins is a dynamic and responsive process, fine-tuned based on specific biotic threats [121]. This adaptive capacity allows plants to mount tailored defenses against the diverse array of BF they encounter, reinforcing the central role of phytoalexins in the evolutionary battle for survival [310].

#### 7.1.3. Activation of Signaling Pathways

Salicylic Acid (SA)

Upon detecting pathogen-associated molecular patterns or infection initiation, plants activate a complex signaling cascade involving SA as a central regulator [311]. SA accumulation triggers the downstream expression of numerous defense-related genes, orchestrating a multifaceted response that includes antimicrobial compound production, cell wall fortification, and programmed cell death to contain fungal invasion [311,312,313].

SA-mediated defense signaling is further refined through crosstalk with other phytohormone pathways, allowing plants to tailor their immune response to specific biotrophic fungal threats [314]. The evolutionary refinement of these signaling mechanisms has equipped plants with a dynamic and robust defense arsenal capable of rapidly perceiving and combating the ever-evolving strategies of their fungal counterparts [21].

Jasmonic Acid (JA)

The co-evolutionary battle between plants and BF has driven the emergence of intricate defense signaling pathways centered around JA [80]. Upon detecting biotrophic fungal pathogens, plants rapidly initiate the biosynthesis and accumulation of JA, a central immune response regulator [238]. JA-mediated signaling triggers the differential expression of diverse defense-related genes [315], orchestrating a defensive strategy that includes antimicrobial compound production [84], cell wall strengthening, and programmed cell death induction to restrict fungal invasion [316].

JA signaling pathways exhibit dynamic crosstalk with other phytohormone-dependent cascades, allowing plants to fine-tune their immune response based on the specific biotrophic fungal threat [59,317]. The evolutionary refinement of these signaling mechanisms has equipped plants with a robust and adaptable defense arsenal capable of rapidly perceiving and combating the ever-evolving strategies of their fungal counterparts [318].

#### 7.1.4. Hypersensitive Response and Programmed Cell Death

Plants have developed a potent defense mechanism known as the hypersensitive response [319]. This rapid and localized form of programmed cell death is triggered upon detecting pathogen-associated molecular patterns or the successful delivery of fungal virulence factors, sacrificing infected cells to halt biotrophic invaders’ progression [320].

The hypersensitive response is underpinned by a complex signaling cascade that mobilizes defense-related gene expression, antimicrobial compound production, and cell wall reinforcement strategies [321]. This coordinated response creates a physical and chemical barrier that can restrict fungal penetration and proliferation, limiting the pathogen’s spread throughout the host plant [322]. The evolutionary refinement of the hypersensitive response has equipped plants with a potent and versatile defense mechanism capable of thwarting diverse biotrophic fungal strategies [323].

#### 7.1.5. Systemic Acquired Resistance (SAR)

Coevolutionary dynamics between plants and BF have driven the development of an adaptive defense strategy known as SAR [324]. This sophisticated signaling network is initiated by local detection of fungal pathogens, triggering the systemic accumulation of SA throughout the plant [229]. SAR coordinates the transcriptional reprogramming of genes involved in antimicrobial compound synthesis, cell wall reinforcement, and priming additional immune responses [229]. The establishment of SAR confers broad-spectrum and long-lasting protection against diverse biotrophic fungal invaders, equipping the plant with a robust defense arsenal [325].

The evolutionary refinement of SAR signaling has endowed plants with the ability to “remember” previous pathogen encounters, allowing for a more rapid and effective defense response upon re-exposure [326]. This adaptive trait, honed through countless cycles of co-evolution, represents a critical mechanism by which plants can anticipate and combat the continually evolving strategies of their fungal counterparts [327].

#### 7.1.6. Evolution of Plant Resistance Genes (NBS-LRR Proteins)

The relentless coevolutionary pressures exerted by BF have driven the rapid evolution and diversification of plant resistance (R) genes encoding nucleotide-binding sites and leucine-rich repeat (NBS-LRR) proteins [328]. These dynamic immune receptors act as sentinels, capable of detecting pathogen-derived effectors and initiating a potent defense response to thwart fungal invasion [329].

The remarkable plasticity of plant genomes has facilitated the proliferation and functional diversification of NBS-LRR genes, with gene duplication, unequal crossing-over, and ectopic recombination events contributing to the expansion and evolution of this critical defense repertoire [330]. This adaptive capacity allows plants to generate novel R gene variants, equipping them with a rapidly evolving arsenal capable of recognizing diverse biotrophic fungal effectors [331]. The evolutionary refinement of NBS-LRR-mediated immunity has thus emerged as a key strategy by which plants maintain a dynamic coexistence with their fungal counterparts [332]. NBS-LRR-mediated immunity constantly adapts to keep pace with the ceaseless innovation of pathogen virulence mechanisms [330].

### 7.2. Strategies of BF to Overcome Plant Defenses

#### 7.2.1. Fungal Effectors That Target and Disrupt Plant Defense Pathways

BF, a unique group of pathogens, have developed sophisticated strategies to overcome the formidable defense mechanisms of their plant hosts [24]. These fungi exhibit a remarkable ability to manipulate and subvert the plant’s immune response, allowing them to thrive and proliferate within the living tissues of their hosts [333].

By secreting effectors, BF can interfere with the plant’s signaling cascades, transcriptional regulation, and enzymatic activities, effectively disarming the plant’s defensive arsenal [55]. By hijacking the plant’s molecular machinery, BF can suppress the activation of defense responses, such as the production of antimicrobial compounds, the strengthening of cell walls, and the initiation of programmed cell death [18]. This delicate balancing act between the fungus and the plant allows the biotrophic pathogen to maintain a harmonious parasitic relationship with its host [334], extracting essential nutrients while avoiding the plant’s lethal counterattacks [335].

#### 7.2.2. Diversification of Fungal Effector Repertoires

The remarkable success of BF in overcoming plant defenses can be largely attributed to the exceptional diversity of their effector repertoires [336]. These fungi have evolved the capacity to produce a vast array of effector molecules, each tailored to target and disrupt specific components of the plant’s defense machinery [27].

Through a combination of gene duplication [337], horizontal gene transfer [338], and rapid evolution, BF have amassed an expansive collection of effectors that can effectively manipulate a wide range of plant defense pathways [56]. This diversification allows the fungi to adapt to the ever-evolving defense mechanisms of their plant hosts, maintaining a competitive advantage and ensuring their continued survival and proliferation [328]. The remarkable plasticity of the effector repertoire enables BF to infect a broader range of plant species [339], expand their ecological niches, and establish complex, long-lasting relationships with their hosts [340], underscoring the intricate coevolutionary dance between these pathogens and their plant counterparts [341].

#### 7.2.3. Adaptive Mechanisms of BF to Evade Plant Immune Detection

BF have evolved sophisticated “stealth” strategies to evade detection and recognition by their plant hosts, often involving lifestyle changes [342]. These fungi have developed the remarkable ability to modify their cellular and molecular profiles during different stages of infection [343], effectively masking their presence and avoiding the triggering of the plant’s robust defense mechanisms [24].

One such strategy employed by BF is the adoption of a stealthy, quiescent lifestyle during the early stages of infection [344]. Instead of immediately manifesting as aggressive pathogens, these fungi may initially colonize the plant’s tissues in a subdued manner, suppressing the expression of virulence factors [345] and maintaining a low metabolic profile [346]. This cloaked presence allows the fungi to establish a foothold within the host without immediately provoking a strong defense response [347].

As the infection progresses, the fungi may then transition to a more active, proliferative stage, deploying a diverse array of effectors to overcome the plant’s defenses and extract the necessary resources for their growth and reproduction [348]. This dynamic, multifaceted lifestyle strategy enables BF to evade plant recognition [121] and successfully exploit their hosts, showcasing the remarkable evolutionary adaptations that have allowed these pathogens to thrive in their respective ecological niches [328].

### 7.3. Co-Evolutionary Adaptations of BF

The co-evolutionary relationship between BF and their plant hosts has given rise to a remarkable array of adaptations on both sides of the pathogenic interaction [348]. These co-evolutionary adaptations have played a pivotal role in shaping the complex dynamics between these organisms, ensuring their continued survival and proliferation [80].

One of the most significant co-evolutionary adaptations observed in BF is their ability to rapidly evolve and diversify their effector repertoires [349]. As plants develop new defense mechanisms to combat fungal invasion, these pathogens respond by generating a vast array of specialized effector molecules that can effectively target and disrupt the plant’s immune pathways [7]. This perpetual “arms race” between the plant and the fungus has driven the evolution of increasingly sophisticated effector strategies [75], allowing BF to maintain a competitive edge and sustain its parasitic lifestyle [350].

Additionally, the development of stealthy colonization strategies, such as the adoption of quiescent growth phases and the formation of specialized feeding structures [21], has enabled BF to evade detection and secure a stable nutritional supply from their plant hosts [15]. These co-evolutionary adaptations, along with the BF’s remarkable capacity for nutrient acquisition [21], have been instrumental in the success of biotrophic plant pathogens, allowing them to thrive in diverse ecological niches and establish complex, long-lasting relationships with their host plants [80].

## 8. Unraveling the Complexity: Known and Emerging Mechanisms in Plant–Fungal Interactions

### 8.1. Pathogen-Associated Molecular Pattern-Triggered Immunity (PTI)

Known mechanisms: Pathogen-associated molecular PTI is a fundamental component of plant innate immunity, playing a critical role in detecting and defending against invading pathogens [351]. The intricate signaling networks and diverse immune responses associated with PTI highlight the importance of continued research [352]. Advancing our understanding of PTI and its interactions with other plant defense mechanisms can lead to novel strategies to enhance crop disease resistance and ensure global food security [353].

Unknown mechanisms and new frontiers: Further studies are required to understand receptor-ligand interactions, including identifying and characterizing additional PRRs that recognize PAMPs from biotrophic fungal pathogens. This involves elucidating the structural basis for PAMP-PRR binding and the molecular mechanisms underlying receptor activation. Additionally, mapping intracellular signaling cascades, such as calcium signaling and MAPK pathways, and exploring crosstalk between PTI and other plant immune pathways, like ETI, is crucial. Understanding plant physiological responses induced by PTI against biotrophic fungal pathogens, including ROS and NOS production, cell wall modifications, and antimicrobial compound synthesis, is essential. Another area of interest is investigating the impact of PTI activation on plant growth, development, and yield under biotrophic fungal pressure.

### 8.2. Effector-Triggered Immunity (ETI)

Known mechanisms: ETI is a key immune response in plants against biotrophic fungal pathogens, induced in response to pathogen effector proteins. These effectors manipulate host cellular processes to promote pathogen replication and transmission. ETI allows plants to distinguish pathogenic from non-pathogenic microbes and mount an appropriate immune response [354]. The molecular mechanisms of ETI are complex and diverse, involving modifications of host target proteins, detection by nucleotide-binding leucine-rich repeat (NLR) proteins, and the use of decoy proteins that mimic host targets to activate NLR sensors [201].

Mechanisms of detection:

Modified Self-Pathogen Effector: ETI recognizes specific pathogen-derived effector proteins modified by the plant’s defense mechanisms [351]. Resistance (R) proteins detect these modifications, triggering a robust immune response [355].

Missing Self-Pathogen Effector: ETI detects the absence or suppression of expected pathogen-derived effector proteins [129].

Stressed Self-Pathogen Effector: ETI recognizes pathogen-derived effector proteins altered by the plant’s defensive stress response [356]. R proteins detect these stress-induced modifications, triggering a defense response [357].

Direct Effector Detection: ETI directly recognizes specific pathogen-derived effector proteins secreted by the fungus [358]. R proteins bind to these effectors, initiating an immune response [163].

Decoy Sensing Effector: ETI uses decoy proteins to mimic the target proteins of pathogen effectors. When effectors bind to decoys, R proteins are activated, triggering the immune response [359].

Unknown mechanisms and new frontiers: Despite significant advances, many aspects of ETI remain unexplored. This includes the precise molecular mechanisms of R protein sensing, the role of epigenetics and small RNA-mediated regulation, and the interplay between ETI and other immune pathways like PTI and systemic acquired resistance. Emerging research is also focusing on engineering novel R protein-effector pairs for broad-spectrum disease resistance.

### 8.3. Danger (or Damage)-Associated Molecular Pattern (DAMP)-Triggered Immunity (DTI)

Known mechanisms: DAMP-triggered immunity (DTI) involves recognizing endogenous molecules released from damaged or stressed plant cells due to pathogen attacks or environmental stresses [201]. DAMPs, such as cell wall fragments and cutin monomers, are detected by plant receptors, triggering immune responses including defense gene activation, antimicrobial compound production, and programmed cell death [360]. DTI provides an additional layer of defense against a wide range of stresses [361,362,363,364].

Unknown mechanisms and new frontiers: The mechanisms underlying DTI are not fully understood, including the diversity of DAMP molecules, specific receptor proteins, and intracellular signaling networks activated upon DAMP perception. The role of epigenetics and small RNAs in DTI and the evolutionary dynamics of DAMP-based defense mechanisms require further exploration. Future research should focus on identifying novel DAMP molecules and receptors, mapping signaling cascades, understanding crosstalk with other defense mechanisms, and exploring epigenetic and small RNA regulation in DTI. Engineering novel DAMP-receptor pairs and exploiting DAMP signaling for crop protection are promising research avenues.

## 9. Conclusions

Our study explored the intricate mechanisms employed by BF to manipulate plant defense responses, focusing on pathways such as JA, SA, ETH, and ROS. Through a comprehensive review of the literature, we have identified various strategies utilized by these fungi to evade or suppress host defenses and facilitate successful colonization. The findings underscore the importance of understanding the sophisticated strategies employed by BF to subvert plant immune responses. By elucidating these mechanisms, researchers can identify potential targets for developing novel strategies to enhance crop resistance against devastating fungal diseases, ultimately contributing to sustainable agriculture and food security. Our study addresses significant gaps in the current understanding of plant–fungus interactions, particularly in the role of hormonal crosstalk and effector proteins in modulating host defense pathways. By synthesizing existing knowledge and highlighting areas for further investigation, we provide a roadmap for future research aimed at unraveling the complexities of biotrophic fungal pathogenesis. While our review provides valuable insights, it is essential to acknowledge certain limitations such as the variability of fungal strains and host-specific responses. Future research should focus on experimental validation of the identified mechanisms in diverse plant–fungus interactions and explore novel approaches, such as omics technologies, to unravel additional layers of complexity. In conclusion, our study sheds light on the sophisticated strategies employed by BF to manipulate plant defense responses, offering valuable insights into host–pathogen interactions. By advancing our understanding of these mechanisms, we pave the way for the development of innovative disease management strategies and contribute to the advancement of agricultural biotechnology.

## Figures and Tables

**Figure 1 jof-10-00635-f001:**
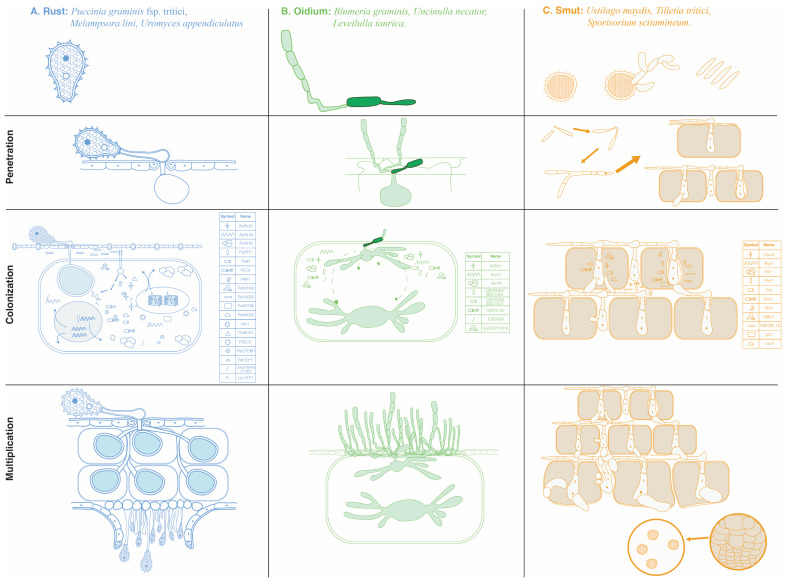
Strategies used by biotrophic fungi in penetration, colonization, and multiplication in living plant cells and tissues. (**A**) Rust fungi (*Puccinia graminis* f. sp. *tritici*, *Melampsora lini*, *Cronartium quercuum*): Penetration: Rust fungi produce appressoria at the tip of a narrow infection peg extending from the germ tube of the fungal spore after landing on the plant surface. Specialized enzymes and mechanical force are used to breach the plant cuticle and cell wall. Colonization: The rust fungus secretes effector proteins (e.g., AvrSr35, AvrSr43, AvrSr45, PgtSR1, Ps87, PEC6, PNPi, Pst02549, Pst18220, Pst03196) from the appressorium, which allow the fungus to evade or suppress the initial plant immune responses and function as a feeding structure within the plant cell. This structure extracts nutrients from the plant cells for fungal growth and reproduction. Multiplication: Rust fungi multiply within the plant leaves, producing new structures and spreading the infection. (**B**) Powdery mildew fungi (*Blumeria graminis*, *Erysiphe necator*, *Leveillula taurica*): Penetration: Oidium spores (conidia) land on the plant surface and germinate, producing a germ tube that differentiates into an appressorium. The penetration peg breaches the plant cell wall mechanically. Colonization: The penetration peg develops into a haustorium, which secretes a broad repertoire of effector proteins (e.g., AVRA1, Avrk1, Avr10, BEC1016, CSEP0064/BEC1054) to silence plant defense responses and facilitate nutrient acquisition. Multiplication: Oidium hyphae produce upright conidiophores bearing chains of conidia that are dispersed by wind to infect new plants. (**C**) Smut fungi (*Ustilago maydis*, *Tilletia indica*, *Sporisorium scitamineum*): Penetration: Smut fungi teliospores germinate to produce haploid yeast-like cells that fuse to form a dikaryotic infection hypha, which develops an appressorium at the tip. Colonization: The infection hypha grows and branches intercellularly within the plant, systemically colonizing the host and secreting effector proteins (e.g., Hum3, Rsp1, Pit2, Pep1). Multiplication: Smut fungi undergo meiosis to produce large numbers of diploid teliospores, which are dispersed by wind or rain. Abbreviation list: Avr (AVR): Avirulence protein; Sr: Stem rust resistance gene. Pgt: *Puccinia graminis* f. sp. *tritici*; SR1: Specific Recognition gene or protein. Ps: *Puccinia striiformis*. PEC: Puccinia effector candidate. PNPi: Puccinia non-pathogenicity inhibitor. Pst: *Puccinia striiformis* f. sp. *tritici*. Shr1: Small secreted protein. SCR: Small Cysteine-Rich protein. PSEC: Puccinia secreted effector candidate. CFEM: Common in Fungal Extracellular Membrane. CEP: Cysteine-rich effector protein. Uaca: *Uromyces appendiculatus* candidate. RTP: Rust-transferred protein. BEC1016: Blumeria Effector Candidate. CSEP: Candidate Secreted Effector Protein. EqCSEP: *Erysiphe quercus* Candidate Secreted Effector Protein. Hum3: *Ustilago maydis* Hydrophobin 3. Rsp1: Receptor-like Secreted Protein 1. Pit2: Protein involved in Translocation 2. Pep1: Protein Essential for Penetration 1. Tin2: Tumor Inducing 2. See1: Seedling Establishment Effector 1. Shy1: Short Hypocotyl 1. CMU1: chorismate mutase 1. ThSCSP_12: Secreted Candidate Small Protein from *Tilletia horrida*. gsr1: Glutathione S-Transferase Related protein 1. uan2: *Ustilago anthracnose*-related protein 2.

**Figure 2 jof-10-00635-f002:**
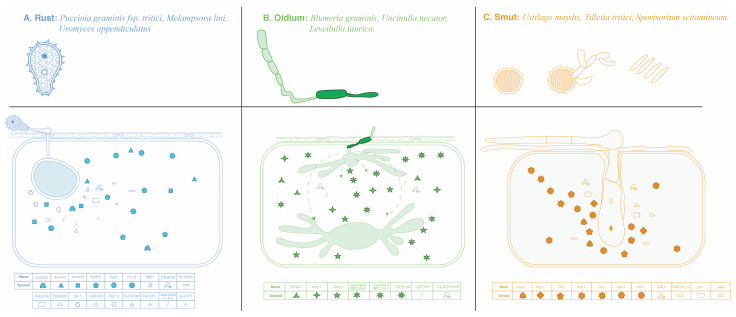
Effector protein repertoire secreted by biotrophic fungi for suppressing plant defense responses. (**A**) Effector secreted by rust fungi (*Puccinia graminis* f. sp. *tritici*, *Melampsora lini*, and *Cronartium quercuum*): AvrSr35, AvrSr43, AvrSr45, PgtSR1, Ps87, PEC6, PNPi, Pst02549, Pst18220, Pst03196, Pst05023, Shr1, PstSCR1, PSEC2, PstCFEM1, PstCEP1, Uaca family, Ua-RTP1, and Ua-RTP1. (**B**) Effector proteins secreted by powdery mildew species (*Blumeria graminis*, *Erysiphe necator*, and *Leveillula taurica*): AVRA1, Avrk1, Avr10, BEC1016, CSEP0064/BEC1054, CSEP0264/BEC1011, CSEP0105, CSEP087, and EqCSEP01276. (**C**) Effector proteins secreted by smut fungi (*Ustilago maydis*, *Tilletia indica*, and *Sporisorium scitamineum*): Hum3, Rsp1, Pit2, Pep1, Tin2, See1, Shy1, CMU1, ThSCSP_12, gsr1, uan2, g2666, g3970, g6610, g1513, g3890, g4549, g1052, g1084, g4554, and g5159.

**Figure 3 jof-10-00635-f003:**
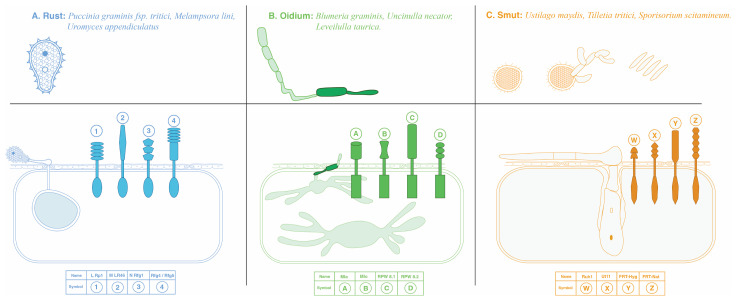
Plant Receptor Proteins (R) involved in the perception of biotrophic pathogenic fungi to induce plant defense responses. (**A**) Plant Receptor Proteins (R) conferring resistance to rust diseases. 1. LRp1: LRR receptor protein 1; 2. MLR46: Mlo-like receptor 46; 3. NRfg1: Nucleotide-binding site leucine-rich repeat receptor 1; 4. Rfg4: Resistance to fungal growth 4. (**B**) Plant Receptor Proteins (R) conferring resistance to powdery mildew diseases. A. Mla: Mildew resistance locus a; B. Mlo: Mildew locus o; C. RPW8.1: Resistance to powdery mildew 8.1; D. RPW8.2: Resistance to powdery mildew 8.2. (**C**) Plant Receptor Proteins (R) conferring resistance to smut diseases. W. Ruh1: Resistance to Ustilago hordei 1; X. Ut11: Ustilago tritici resistance 11; Y. FRT-Hyg: Flanking region transformation with Hygromycin resistance; Z. FRT-Nat: Flanking region transformation with Nourseothricin resistance.

**Table 1 jof-10-00635-t001:** Examples of effector molecules of biotrophic fungi.

Biotrophic Fungi	Effector Molecules	References
(rust fungi)		
*Puccinia graminis*	AvrSr35, AvrSr43, AvrSr45, PgtSR1	[85,86,87,88]
*Puccinia striiformis*	Ps87, PEC6, PNPi, Pst02549, Pst18220, Pst03196, Pst05023, Shr1, PstSCR1, PSEC2, PstCFEM1, PstCEP1	[62,81,89,90,91,92,93,94,95,96,97]
*Uromyces appendiculatus*	Uaca family (1–46), Ua-RTP1	[98,99]
(powdery mildew fungi)		
*Blumeria graminis*	AVRA1, Avrk1, Avr10, BEC1016, CSEP0064/BEC1054, CSEP0264/BEC1011, CSEP0105	[77,100,101,102,103,104]
*Uncinula necator*	CSEP087, EqCSEP01276	[105,106]
(smut fungi)		
*Ustilago maydis*	Hum3, Rsp1, Pit2, Pep1, Tin2, See1, Shy1, CMU1	[16,107,108,109,110,111,112,113]
*Tilletia horrida*	ThSCSP_12, gsr1, uan2	[114,115]
*Sporisorium scitaminea*	g2666, g3970, g6610, g1513, g3890, g4549, g1052, g4554, g5159	[116,117]

**Table 2 jof-10-00635-t002:** Summary of genes and proteins involved in plant defense responses against various biotrophic fungi (BF).

Gene/Protein	Type	Plant Species	BF	Function	Reference
RLP1	RLP	*Zea mays*	*Ustilago maydis*	Targeted by effector Tin2, leading to degradation and suppression of defense responses	[111]
Rp1, Rp3	RLP	*Zea mays*, *Triticum aestivum*	*Rust fungi*	Recognizes specific avirulence effectors produced by rust fungi	[136]
AvrL567	Effector	*Linum usitatissimum*	*Melampsora lini*	Targets transcription factor TaSPT6	[137]
Lr10	LRR-RLK	*Triticum aestivum*	*Puccinia graminis* f. sp. *tritici*	Confers resistance by recognizing pathogen-derived molecules	[138]
Lr21	LRR-RLK	*Triticum aestivum*	*Puccinia triticina*	Confers resistance by recognizing specific avirulence effectors	[139]
Sr33	LRR-RLK	*Triticum aestivum*	*Puccinia graminis* f. sp. *tritici*	Confers resistance by recognizing AvrSr33 effector	[140]
Rpg1	LRR-RLK	*Hordeum vulgare*	*Puccinia graminis* f. sp. *hordei*	Confers resistance by recognizing specific avirulence effectors	[141]
REN1	LRR-RLK	*Vitis vinifera*	*Erysiphe necator*	Confers resistance by recognizing specific avirulence effectors	[142]
PMR4	LRR-RLK	*Arabidopsis thaliana*	*Golovinomyces cichoracearum*	Confers resistance by recognizing specific pathogen-derived molecules	[143]
Mla	LRR-RLK	*Hordeum vulgare*	*Blumeria graminis* f. sp. *hordei*	Confers resistance by recognizing specific Avr proteins	[144]
Pm3	LRR-RLK	*Triticum aestivum*	*Blumeria graminis* f. sp. *tritici*	Confers resistance by recognizing Avr proteins	[145]
Mlo	LRR-RLK	*Hordeum vulgare*	*Blumeria graminis* f. sp. *hordei*	Confers broad-spectrum resistance when mutated	[100]
Sr33	NLR	*Triticum aestivum*	*Puccinia graminis* f. sp. *tritici*	Confers resistance by recognizing AvrSr33 effector	[140]
Sr35	NLR	*Triticum aestivum*	*Puccinia graminis* f. sp. *tritici*	Confers resistance by recognizing AvrSr35 effector	[146]
Rpg1	NLR	*Hordeum vulgare*	*Puccinia graminis* f. sp. *hordei*	Confers resistance by recognizing AvrRpg1 effector	[147]
Rpp1	NLR	*Glycine max*	*Phakopsora pachyrhizi*	Confers resistance by recognizing AvrRpp1 effector	[148]
RPW8-NLR	NLR	*Arabidopsis thaliana*	*Oidium* spp.	Confers resistance against powdery mildew	[149]
MLA-NLR	NLR	*Hordeum vulgare*	*Blumeria graminis* f. sp. *hordei*	Confers resistance by recognizing specific Avr effectors	[150]
PMR4-NLR	NLR	*Arabidopsis thaliana*	*Oidium* spp.	Confers resistance by triggering defense responses	[151]
SERK3, SERK1	SERK	*Solanum lycopersicum*	*Cladsporium fulvum*	Involved in recognition of Avr9 effector	[152]
TaWRKY1	WRKY	*Triticum aestivum*	*Puccinia striiformis* f. sp. *tritici*	Targeted by AvrStb6, triggering ETI responses against *Zymoseptoria tritici*	[153]
TaSYP71	Syntaxin	*Triticum aestivum*	*Blumeria graminis* f. sp. *tritici*	Involved in vesicle trafficking and defense responses, targeted by AvrPm3b	[154]

Note: RLP: Receptor-Like Protein; LRR-RLK: Leucine-Rich Repeat Receptor-Like Kinase; NLR: Nucleotide-Binding Leucine-Rich Repeat; SERK: Somatic Embryogenesis Receptor Kinase; WRKY: WRKY Transcription Factor; Syntaxin: Syntaxin Protein; PMR4: Powdery Mildew Resistance 4; REN1: Resistance to *Erysiphe necator* 1; Mlo: Mildew resistance locus.

**Table 3 jof-10-00635-t003:** Mechanisms of effector-mediated suppression of plant defense responses.

Mechanism	Description	Details	References
1. Disruption of defense signaling pathways	Effectors disrupt the defense signaling pathways of host plants	Effectors target key signaling components, leading to compromised defense responses.	[162]
1.1. Interference with downstream signaling components involved in defense responses	Effectors interfere with the activation of defense-related genes or signaling cascades.	Interference with activation of defense-related genes or signaling cascades	[53]
2. Mimic or block recognition of PAMPs	Effectors may mimic or block the recognition of pathogen-associated molecular patterns (PAMPs) by pattern recognition receptors (PRRs).	Effector molecules mimic PAMPs or block PRR binding sites.	[163]
3. Interaction with host proteins	Effector molecules interact with specific host proteins such as receptor-like kinases (RLKs) and transcription factors.	Interaction with RLKs disrupts defense signal perception; Interaction with transcription factors manipulates gene expression.	[164,165]

**Table 4 jof-10-00635-t004:** Strategies used by biotrophic fungi in penetration, colonization, and multiplication in living plant cells and tissues.

Fungal Group	Example Species	Penetration Mechanism	Key Effectors	Nutrient Acquisition Mechanism	Reproduction Strategy
Rust Fungi	*Puccinia graminis* f. sp. *tritici*, *Melampsora lini*, *Uromyces appendiculatus*	Appressoria formation at the tip of infection pegs, breaching cuticle and cell wall using enzymes and mechanical force	AvrSr35, AvrSr43, AvrSr45, PgtSR1, Ps87, PEC6, PNPi, Pst02549, Pst18220, Pst03196, Pst05023, Shr1, PstSCR1, PSEC2, PstCFEM1, PstCEP1, Uaca family (1–46), Ua-RTP1, Ua-RTP1	Haustorium formation within plant cell	Growth and reproduction through extraction of nutrients from plant cells, multiplication of new fungal structures
Powdery mildew fungi	*Blumeria graminis*, *Uncinula necator*, *Leveilulla taurica*	Conidia germination, germ tube differentiation into appressoria, mechanical breaching of the cell wall	AVRA1, Avrk1, Avr10, BEC1016, CSEP0064/BEC1054, CSEP0264/BEC1011, CSEP0105, CSEP087, EqCSEP01276	Haustorium secretion of protein effectors to silence plant defenses	Rapid colonization and infection through asexual spores called conidia, dispersed by wind
Smut Fungi	*Ustilago maydis*, *Tilletia tritici*, *Sporisorium scitamineum*	Teliospore germination, formation of dikaryotic infection hyphae, appressorium development at hyphae tips	Hum3, Rsp1, Pit2, Pep1, Tin2, See1, Shy1, CMU1, ThSCSP_12, gsr1, uan2, g2666, g3970, g6610, g1513, g3890, g4549, g1052, g1084, g4554, g5159	Systemic colonization via intercellular hyphae, nutrient extraction from plant tissues	Systemic colonization, followed by meiosis and production of diploid teliospores, dispersed by wind or rain

**Table 5 jof-10-00635-t005:** Mechanisms of effector-triggered immunity and gene-for-gene interactions in plant–pathogen interactions.

Mechanism	Description	Details	References
Effector-Triggered Immunity (ETI)	Plants respond to pathogenic microorganisms by detecting specific effectors secreted by the pathogens.	Recognition of effectors by plant resistance proteins (R proteins) leads to activation of a signaling cascade and subsequent defense responses such as hypersensitive response (HR), production of reactive oxygen species (ROS), and antimicrobial compounds.	[191]
Gene-for-Gene Interaction	Each R protein recognizes a corresponding effector molecule produced by the pathogen.	Recognition of specific effectors secreted by *Puccinia graminis* by R genes such as Sr33, Sr35, Sr39, Sr21	[192]
	ETI against *P. striiformis* involves the recognition of specific effectors by R proteins in wheat.	R genes Yr5, Yr10, Yr15, and Yr17 confer resistance against *P. striiformis*.	[81]
	Recognition of specific effectors by R genes in barley and wheat leading to resistance against powdery mildew.	Mla and Mlo genes in barley; PMR (Powdery mildew resistance genes) in wheat	[193,194,195]
	ETI against *U. maydis* involves the recognition of specific effectors by plant R genes.	Rp1-D, Rp3, Rp6 in maize conferring resistance against rust diseases.	[196]

**Table 6 jof-10-00635-t006:** Plant defense mechanisms and corresponding R genes against biotrophic fungi.

Biotrophic Fungus	Plant Defense	R Genes	References
(rust fungi)			
*Puccinia graminis*	LRR-RLK	Lr10, Lr21	[197]
*Puccinia graminis*	LRR-RLK	Sr22, Sr33, Sr35	[157]
*Puccinia graminis*	NLRs	Rpg1	[198]
(powdery mildew fungi)			
*Uncinula necator*	LRR-RLK	REN1	[199]
*Oidium neolycopersici*	LRR-RLK	PMR4	[200]
*Blumeria graminis*	LRR-RLK	Mla	[185]
*Blumeria graminis*	LRR-RLK	Pm3	[201]
*Blumeria graminis*	LRR-RLK	Mlo	[202]
(smut fungi)			
*Ustilago maydis*	RLPs	Rp1 and Rp3	[203]

**Table 7 jof-10-00635-t007:** The role of plant hormones in suppressing defense responses against biotrophic fungi (BF).

Plant Hormones	Mechanism	Details	References
Jasmonic Acid (JA)	Manipulation of JA pathway by secreting effector molecules, interfering with JA signaling, and suppressing activation of defense responses	Inhibition of defense-related genes and production and secondary metabolites	[210,272]
	Interfering with JA biosynthesis or perception, promoting JA accumulation, and suppressing SA	Evasion of plant immune responses by suppressing JA-mediated defense pathways	[238,273]
Salicylic Acid (SA)	Direct targeting of SA pathway components, inhibiting SA biosynthesis, perception, or signaling	Inactivation of defense responses, including hypersensitive response (HR) and systemic acquired resistance (SAR)	[59,274,275]
	Promotion of JA or ETH synthesis while suppressing SA, shifting defense response away from SA-mediated defenses	Inhibition of SA-mediated defenses, facilitating colonization by BF	[274]
Abscisic Acid (ABA)	Suppression of ABA synthesis	Inhibition of ABA biosynthesis pathways, reducing ABA levels	[252]
	Interference with ABA signaling pathways	Manipulation of ABA receptors or downstream signaling components	[276]
	Manipulation of hormonal balance	Modulation of the balance between ABA and other hormones such as JA, SA, and ETH	[231]
Ethylene (ETH)	Disruption of ETH signaling pathway, inhibition of ETH biosynthesis or perception, leading to reduced ETH production	Induction of ETH-responsive genes involved in defense responses against BF	[59]

**Table 8 jof-10-00635-t008:** Biotrophic fungi strategies for suppressing plant defense responses against ROS.

Biotrophic Fungus	Mechanisms	Examples	References
*Puccinia graminis*	Production of antioxidant enzymes (SOD, CAT, POX) to neutralize ROS, secretion of effector molecules to interfere with ROS production or signaling pathways	SOD, CAT, POX, effector molecules	[295]
*Phakopsora pachyrhizi*	Secretion of effector molecules to potentially manipulate host plant defenses, including modulation of ROS production or signaling pathways	Effector molecules	[296]
*Ustilago maydis*	Production of antioxidant enzymes (SOD, CAT, POX), synthesis of glutathione, melanin pigments with antioxidant properties	SOD, CAT, POX, glutathione, melanin	[297]

## Data Availability

Details of data availability are available from the first author on request.

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
