# Peer review of "Uncovering the Mechanisms: The Role of Biotrophic Fungi in Activating or Suppressing Plant Defense Responses"

_jof, 2024, doi:10.3390/jof10090635_

Round 1

Reviewer 1 Report

The authors provide a thorough examination of the strategies employed by biotrophic fungi to manipulate plant defense responses. The review is well-structured, with clear subsections that explore many mechanisms utilized by these fungi, including hormone manipulation, inhibition of programmed cell death, effector-mediated interference, detoxification of reactive oxygen species, and modulation of nitric oxide signaling pathways. Additionally, the manuscript discusses how biotrophic fungi effector proteins interact with key host proteins, such as receptor-like kinases and transcription factors, to promote fungal growth while suppressing plant defense responses.

While the review provides a detailed examination of the mechanisms employed by BF to suppress plant defense responses, it could benefit from a more balanced discussion of both known and unknown mechanisms. Including potential areas for future research or unresolved questions in such a field would enrich the discussions and provide direction for future studies. Authors may want to provide more of their own opinions on the topics.

Author Response

For review article:

Manuscript ID jof-3063704: Uncovering the Mechanisms: The Role of Biotrophic Fungi in Activating or Suppressing Plant Defense Responses

Response to Reviewer 1 Comments

  1. Summary: Thank you very much for taking the time to review this manuscript. Please find the detailed responses below and the corresponding revisions/corrections highlighted/in track changes in the re-submitted files.
  2. Questions for General Evaluation

    Reviewer’s Evaluation

    Is the work a significant contribution to the field?

    YES

    Is the work well organized and comprehensively described?

    YES

    Is the work scientifically sound and not misleading?

    YES

    Are there appropriate and adequate references to related and previous work?          

    YES

    Is the English used correct and readable?      

    YES

  3. Point-by-point response to Comments and Suggestions for Authors: Major comments: The authors provide a thorough examination of the strategies employed by biotrophic fungi to manipulate plant defense responses. The review is well-structured, with clear subsections that explore many mechanisms utilized by these fungi, including hormone manipulation, inhibition of programmed cell death, effector-mediated interference, detoxification of reactive oxygen species, and modulation of nitric oxide signaling pathways. Additionally, the manuscript discusses how biotrophic fungi effector proteins interact with key host proteins, such as receptor-like kinases and transcription factors, to promote fungal growth while suppressing plant defense responses. Response of major comments: Thank you for thoroughly evaluating our manuscript. We are grateful for your insightful comments and are pleased that you found our review well-structured and comprehensive.

    Comments 1. While the review provides a detailed examination of the mechanisms employed by BF to suppress plant defense responses, it could benefit from a more balanced discussion of both known and unknown mechanisms. Including potential areas for future research or unresolved questions in such a field would enrich the discussions and provide direction for future studies. Authors may want to provide more of their own opinions on the topics.

    Response 1. Thank you for your detailed and constructive comments on our manuscript foe review paper. We appreciate your feedback and are committed to addressing your suggestions to improve the quality and impact of our review. Below, we respond to your specific comments:

    We recognize the importance of offering a balanced discussion of both the known and unknown mechanisms that biotrophic fungi use to manipulate plant defense responses. To address this, we have added a new Section 8 in the revised document (see LNs: 831-899):

    1. Unraveling the Complexity: Known and Emerging Mechanisms in Plant-Fungal Interactions.

    8.1. Pathogen-Associated Molecular Pattern-Triggered Immunity (PTI)

    Known mechanisms: Pathogen-associated molecular PTI is a fundamental component of plant innate immunity, playing a critical role in detecting and defending against invading pathogens [351]. The intricate signaling networks and diverse immune responses associated with PTI highlight the importance of continued research [352]. Advancing our understanding of PTI and its interactions with other plant defense mechanisms can lead to novel strategies to enhance crop disease resistance and ensure global food security [353].

    Unknown mechanisms and new frontiers: Further studies are required to understand receptor-ligand interactions, including identifying and characterizing additional PRRs that recognize PAMPs from biotrophic fungal pathogens. This involves elucidating the structural basis for PAMP-PRR binding and the molecular mechanisms underlying receptor activation. Additionally, mapping intracellular signaling cascades, such as calcium signaling and MAPK pathways, and exploring crosstalk between PTI and other plant immune pathways, like ETI, is crucial. Understanding plant physiological responses induced by PTI against biotrophic fungal pathogens, including ROS and NOS production, cell wall modifications, and antimicrobial compound synthesis, is essential. Another area of interest is investigating the impact of PTI activation on plant growth, development, and yield under biotrophic fungal pressure.

    8.2. Effector-Triggered Immunity (ETI)

    Known mechanisms: ETI is a key immune response in plants against biotrophic fungal pathogens, induced in response to pathogen effector proteins. These effectors manipulate host cellular processes to promote pathogen replication and transmission. ETI allows plants to distinguish pathogenic from non-pathogenic microbes and mount an appropriate immune response [354]. The molecular mechanisms of ETI are complex and diverse, involving modifications of host target proteins, detection by nucleotide-binding leucine-rich repeat (NLR) proteins, and the use of decoy proteins that mimic host targets to activate NLR sensors [355].

    Mechanisms of detection:

    Modified Self-Pathogen Effector: ETI recognizes specific pathogen-derived effector proteins modified by the plant's defense mechanisms [351]. Resistance (R) proteins detect these modifications, triggering a robust immune response [356].

    Missing Self-Pathogen Effector: ETI detects the absence or suppression of expected pathogen-derived effector proteins [129].

    Stressed Self-Pathogen Effector: ETI recognizes pathogen-derived effector proteins altered by the plant's defensive stress response [357]. R proteins detect these stress-induced modifications, triggering a defense response [358].

    Direct Effector Detection: ETI directly recognizes specific pathogen-derived effector proteins secreted by the fungus [359]. R proteins bind to these effectors, initiating an immune response [163].

    Decoy Sensing Effector: ETI uses decoy proteins to mimic the target proteins of pathogen effectors. When effectors bind to decoys, R proteins are activated, triggering the immune response [360].

    Unknown mechanisms and new frontiers: Despite significant advances, many aspects of ETI remain unexplored. This includes the precise molecular mechanisms of R protein sensing, the role of epigenetics and small RNA-mediated regulation, and the interplay between ETI and other immune pathways like PTI and systemic acquired resistance. Emerging research is also focusing on engineering novel R protein-effector pairs for broad-spectrum disease resistance.

    8.3. Danger (or Damage)-Associated Molecular Pattern (DAMP)-Triggered Immunity (DTI)

    Known mechanisms: DAMP-triggered immunity (DTI) involves recognizing endogenous molecules released from damaged or stressed plant cells due to pathogen attacks or environmental stresses [355]. DAMPs, such as cell wall fragments and cutin monomers, are detected by plant receptors, triggering immune responses including defense gene activation, antimicrobial compound production, and programmed cell death [361]. DTI provides an additional layer of defense against a wide range of stresses [362].

    Unknown mechanisms and new frontiers: The mechanisms underlying DTI are not fully understood, including the diversity of DAMP molecules, specific receptor proteins, and intracellular signaling networks activated upon DAMP perception. The role of epigenetics and small RNAs in DTI, and the evolutionary dynamics of DAMP-based defense mechanisms, require further exploration. Future research should focus on identifying novel DAMP molecules and receptors, mapping signaling cascades, understanding crosstalk with other defense mechanisms, and exploring epigenetic and small RNA regulation in DTI. Engineering novel DAMP-receptor pairs and exploiting DAMP signaling for crop protection are promising research avenues.

Reviewer 2 Report

This review examines recent data on the interaction of biotrophic fungi with plants, namely how these pathogens are able to activate or suppress the plant defence response. The idea of the manuscript is interesting and has practical relevance in agriculture. Despite the large number of reviews on this topic, this manuscript has collected unique data, focusing on the protective pathways of plants against biotrophic fungi involving phytohormones jasmonic acid (JA), salicylic acid (SA), abscisic acid (ABA) as well as reactive oxygen species (ROS). The manuscript is generally well-written and has good illustrations. However, authors should check the text carefully, as there are often repetitions in the meaning. I recommend this Ms for publication after essential major revisions.

1.                  Authors should rewrite the beginning of the introduction because the meaning of the sentences is repeated. For example, the meaning of the sentence on line 34 repeats the meaning of the sentence on lines 38-39. The sentences on lines 36-37 are repeated with the sentence on lines 42-43.

2.                  It is often mentioned in the text that biographical fungi secrete effector molecules. However, the authors do not specify which ones. I think the authors should give examples of effector molecules, what kind of molecules they are, what they are called.

Lines 90, 227, 376, 408, 447, 499, 539, 553 – it is necessary to specify which specific effectors.

3.                  The information on lines 124-216 should be presented in a table similar to Table 4.

4.                  The sentences on lines 224-227 and lines 230-231 have the same semantic meaning, they should be rewritten or combined.

5.                  Figure 1. In the first column for rust, the colonisation row shows a diagram of the molecular interactions that occur in a plant cell when a fungus invades. a) decipher the abbreviations in the figure legend, which are shown in the figure; b) why are there no similar schemes for Oidium and Smut?

6.                  The information on lines 290-327 should also be presented in a table similar to Table 4.

7.                  Figure 2. In the caption to the figure, decipher the names of the proteins shown schematically in the figure.

8.                  Figure 3. What do the symbols 1-4, A-D, W-Z mean? It is necessary to specify the names in the caption of the figure.

9.                  The sentences on lines 397-398 and lines 451-454 have the same semantic meaning, they should be rewritten or deleted.

10.              The information on lines 473-493 should also be presented in a table similar to Table 4.

11.              The sentence on line 510 should be moved to the beginning of the chapter.

12.              Perhaps another chapter should be added describing the evolutionary aspect of the relationship between biotrophic fungi and plants. How pathogens have learned to activate or reduce the plant's defence response.

Minor Essential Revisions:

Line 450: Puccinia – italicize;

Line 553: U. maydis – italicize;

Author Response

For review article:

Manuscript ID jof-3063704: Uncovering the Mechanisms: The Role of Biotrophic Fungi in Activating or Suppressing Plant Defense Responses

Response to Reviewer 2 Comments

  1. Summary: Thank you very much for taking the time to review this manuscript. Please find the detailed responses below and the corresponding revisions/corrections highlighted/in track changes in the re-submitted files.
  1.  

Questions for General Evaluation

Reviewer’s Evaluation

Is the work a significant contribution to the field?

YES

Is the work well organized and comprehensively described?

YES

Is the work scientifically sound and not misleading?

YES

Are there appropriate and adequate references to related and previous work?          

YES

Is the English used correct and readable?      

Minor editing of English language required.

  1. Point-by-point response to Comments and Suggestions for Authors: Major comments: This review examines recent data on the interaction of biotrophic fungi with plants, namely how these pathogens can activate or suppress the plant defense response. The idea of the manuscript is interesting and has practical relevance in agriculture. Despite the large number of reviews on this topic, this manuscript has collected unique data, focusing on the protective pathways of plants against biotrophic fungi involving phytohormones ‒ jasmonic acid (JA), salicylic acid (SA), abscisic acid (ABA) ‒ as well as reactive oxygen species (ROS). The manuscript is generally well-written and has good illustrations. However, authors should check the text carefully, as there are often repetitions in the meaning. I recommend this Ms for publication after essential major revisions. Response of major comments: The authors sincerely appreciate the Reviewer's major comments thorough review and valuable feedback on our manuscript. We are particularly grateful for your positive comments regarding the manuscript's relevance, the uniqueness of the data presented, and the quality of the illustrations. Regarding your observation about the repetition of meanings within the text, we want you to know about this concern and have carefully revised the manuscript to avoid redundancies. We have restructured several sections to improve clarity and coherence, ensuring that each point is articulated distinctly. We hope these revisions meet your expectations, and we are confident that the manuscript is now better suited for publication. We remain open to any further suggestions you may have to improve the manuscript further. Thank you again for your insightful comments and for recommending our manuscript for publication after major revisions.

Reply to Detailed comments

Comments 1. Authors should rewrite the beginning of the introduction because the meaning of the sentences is repeated. For example, the meaning of the sentence on line 34 repeats the meaning of the sentence on lines 38-39. The sentences on lines 36-37 are repeated with the sentence on lines 42-43.

Response 1. Thank you for your valuable feedback on our manuscript, particularly regarding the repetition of meaning in the introduction. We greatly appreciate your thorough review and agree that the clarity and distinctiveness of ideas in the introduction are essential. In response to your comments, we have not only addressed the specific repetitions you mentioned but have also taken the opportunity to rewrite the entire introduction section (please, read the LNs 35-85). This comprehensive revision allowed us to enhance the structure and flow of the content. The revised introduction now begins with an overview of biotrophic fungi, including rusts, powdery mildews, and smuts, and their impact on agriculture. It is followed by detailed subsections on rust fungi, powdery mildew fungi, and smut fungi, highlighting their biological characteristics, host interactions, and strategies for evading plant defense mechanisms. We believe these revisions have significantly improved the clarity and coherence of the manuscript, ensuring that each point is clearly articulated without repetition.

Comments 2. It is often mentioned in the text that biographical fungi secrete effector molecules. However, the authors do not specify which ones. I think the authors should give examples of effector molecules, what kind of molecules they are, what they are called. Lines 90, 227, 376, 408, 447, 499, 539, 553 – it is necessary to specify which specific effectors.

Response 2: Thank you for your valuable feedback on our manuscript. We appreciate your suggestion to provide specific examples of effector molecules secreted by biotrophic fungi, as this will significantly enhance the depth and clarity of our discussion. In response to your comments, we have thoroughly revised the relevant sections of the manuscript. Specifically, we have rewritten the entire "Mechanisms of BF Impact on Plant Hosts" section to include detailed descriptions of the effector molecules secreted by various biotrophic fungi (please, read the LNs 86-220). Furthermore, we have created a comprehensive Table 1 that lists notable examples of effector molecules produced by various biotrophic fungi, along with their references. This addition ensures that readers have a clear understanding of the specific molecules involved in the interactions discussed throughout the manuscript. We believe that these revisions have significantly improved the manuscript's clarity and the depth of information presented, addressing your concerns effectively. Thank you again for your constructive comments. We hope that the revisions meet your expectations.

Line 90; now appeared this information in Table 2.

Line 227; now appeared in Table 2.

Line 376; now appeared in LN 436-439 as: Ustilago maydis: ETI against U. maydis involves the recognition of specific effectors by plant R genes, leading to robust immune responses. Examples include Rp1-D, Rp3, and Rp6 in maize by recognizing specific effectors such as Pep1 and Cmu1 and activating defense responses [190].

Line 408; now appeared in LN 590,591 as: Blumeria graminis, the causal agent of powdery mildew disease, primarily suppresses the SA pathway (Cmu1) rather than disrupting it.  

Line 499; now appeared in LNs 605-607 as: U. maydis secretes enzymes, such as salicylate hydroxylase, and molecules like ZmCm2 and Cmu1, that directly inhibit enzymes involved in the SA biosynthesis path-way [247].

Line 539; now appeared in LN 658-660 as: Several mechanisms have been proposed for this interference: i) production of effector proteins (e.g., Jsi1 from Ustilago maydis) that target and inhibit key enzymes involved in ETH biosynthesis, resulting in reduced ETH production [247],

Line 553; now appeared in LNs 655-658 as: It produces effectors such as TuMYB46L and TuACO3, which regulate ETH biosynthesis in wheat, impacting defense responses associated with ETH signaling.

Comments 3. The information on lines 124-216 should be presented in a table similar to Table 4.

Response 3: Thank you for your constructive feedback on our manuscript. We appreciate your suggestion to present the information in lines 124-216 in a tabular format similar to Table 4. In response, we have reorganized this section and incorporated additional data from new studies into a table. We believe this revision enhances the clarity and accessibility of the information, allowing readers to more easily compare and understand the key points. This change also ensures consistency with the other tables in the manuscript, improving the overall presentation. We hope that this revision meets your expectations and strengthens the manuscript. Thank you again for your valuable insights.

Please, read the LNs 274-289 where Table 2 provides a comprehensive summary of various genes and proteins involved in plant defense responses against a range of pathogens. It highlights different types of proteins such as RLPs (Receptor-Like Proteins), LRR-RLKs (Leucine-Rich Repeat Receptor-Like Kinases), NLRs (Nucleotide-Binding Leucine-Rich Repeat Receptors), SERKs (Somatic Embryogenesis Receptor Kinases), WRKY transcription factors, and syntaxins. Each entry details the specific plant species in which these proteins are found, the corresponding pathogen they defend against, and their role in the plant's immune response. The table underscores the complex interplay between plant immune receptors and pathogen effectors, illustrating how plants recognize and respond to pathogenic threats to trigger effective defense mechanisms. This detailed information serves as a valuable resource for understanding plant-pathogen interactions and the molecular basis of plant immunity.

Response 4: Thank you for pointing out the redundancy between the sentences on lines 224-227 and 230-231. We have carefully reviewed these statements and agree that they convey similar meanings. To address this, we have rewritten the sentences to eliminate redundancy while preserving the intended information in Sub-section 2.2. Secrete Effector Proteins in Plant (LNs 126-220). This revision improves the clarity and flow of the text. We appreciate your attention to detail and believe this change strengthens the manuscript.

 Comments 5: Figure 1. In the first column for rust, the colonization row shows a diagram of the molecular interactions that occur in a plant cell when a fungus invades. a) decipher the abbreviations in the figure legend, which are shown in the figure; b) why are there no similar schemes for Oidium and Smut?

Response 5: Thank you for your insightful feedback regarding Figure 1. We have carefully considered your suggestions and made the necessary adjustments.

Figure 1. Strategies used by biotrophic fungi in penetration, colonization, and multiplication in living plant cells and tissues.

Point a: Deciphering Abbreviations

Thank you for pointing out the need to decipher the abbreviations in the legend of Figure 1. We have updated the figure legend to include a detailed explanation of the abbreviations used, along with a comprehensive description of the effector proteins involved. Below is the revised legend for Figure 1, which now clearly defines each abbreviation and provides a detailed explanation of the strategies employed by rust fungi, powdery mildew fungi, and smut fungi during the stages of penetration, colonization, and multiplication within plant cells.

  1. Rust Fungi (Puccinia graminis f.sp. tritici, Melampsora lini, Cronartium quercuum):

    Penetration: Rust fungi produce appressoria at the tip of a narrow infection peg, extending from the germ tube of the fungal spore after landing on the plant surface. Specialized enzymes and mechanical force are used to breach the plant cuticle and cell wall. Colonization: The rust fungus secretes effector proteins (e.g., AvrSr35, AvrSr43, AvrSr45, PgtSR1, Ps87, PEC6, PNPi, Pst02549, Pst18220, Pst03196, Pst05023, Shr1, PstSCR1, PSEC2, PstCFEM1, PstCEP1, Uaca family (1-46), Ua-RTP1, Ua-RTP2) from the appressorium, which allow the fungus to evade or suppress the initial plant immune responses and function as a feeding structure within the plant cell. This structure extracts nutrients from the plant cells for fungal growth and reproduction. Multiplication: Rust fungi multiply within the plant, producing new structures and spreading the infection.

  1. Powdery Mildew Fungi (Blumeria graminis, Erysiphe necator, Leveillula taurica):

    Penetration: Oidium spores (conidia) land on the plant surface and germinate, producing a germ tube that differentiates into an appressorium. The penetration peg breaches the plant cell wall mechanically. Colonization: The penetration peg develops into a haustorium, which secretes a broad repertoire of effector proteins (e.g., AVRA1, Avrk1, Avr10, BEC1016, CSEP0064/BEC1054, EqCSEP01276) to silence plant defense responses and facilitate nutrient acquisition. Multiplication: Oidium hyphae produce upright conidiophores bearing chains of conidia that are dispersed by wind to infect new plants.

  1. Smut Fungi (Ustilago maydis, Tilletia indica, Sporisorium scitamineum):

    Penetration: Smut fungi teliospores germinate to produce haploid yeast-like cells that fuse to form a dikaryotic infection hypha, which develops an appressorium at the tip. Colonization: The infection hypha grows and branches intercellularly within the plant, systemically colonizing the host and secreting effector proteins (e.g., Hum3, Rsp1, Pit2, Pep1, Tin2, See1, Shy1, CMU1, ThSCSP_12, gsr1, uan2). Multiplication: Smut fungi undergo meiosis to produce large numbers of diploid teliospores, which are dispersed by wind or rain.

Abbreviation list: Avr (AVR): Avirulence protein; Sr: Stem rust resistance gene. Pgt: Puccinia graminis f.sp. tritici; SR1: Specific Recognition gene or protein. Ps: Puccinia striiformis. PEC: Puccinia effector candidate. PNPi: Puccinia non-pathogenicity inhibitor. Pst: Puccinia striiformis f.sp. tritici. Shr1: Small secreted protein. SCR: Small Cysteine-Rich protein. PSEC: Puccinia secreted effector candidate. CFEM: Common in Fungal Extracellular Membrane. CEP: Cysteine-rich effector protein. Uaca: Uromyces appendiculatus candidate. RTP: Rust-transferred protein. BEC1016: Blumeria Effector Candidate. CSEP: Candidate Secreted Effector Protein. EqCSEP: Erysiphe quercus Candidate Secreted Effector Protein. Hum3: Ustilago maydis Hydrophobin 3. Rsp1: Receptor-like Secreted Protein 1. Pit2: Protein involved in Translocation 2. Pep1: Protein Essential for Penetration 1. Tin2: Tumor Inducing 2. See1: Seedling Establishment Effector 1. Shy1: Short Hypocotyl 1. CMU1: Carbon-Metabolism-Related Ustilago 1. ThSCSP_12: Secreted Candidate Small Protein from Tilletia horrida. gsr1: Glutathione S-Transferase Related protein 1. uan2: Ustilago anthracnose-related protein 2.

b) Lack of Molecular Interaction Schemes for Oidium and Smut:

We appreciate your observation regarding the absence of detailed molecular interaction schemes for Oidium and Smut fungi in the original figure. In response, we have updated Figure 1 to include comprehensive molecular interaction schemes for Oidium and Smut fungi, similar to those provided for rust fungi. Additionally, we have focused on the overall strategies used by Oidium and Smut fungi in penetration, colonization, and multiplication, as detailed in the legend. This approach ensures that the figure remains accurate and is grounded in the current understanding of these pathogens, reflecting the nuanced and well-understood nature of their interactions. The updated Figure 1, now aligns with the current understanding of the molecular interactions for these pathogens, ensuring consistency and clarity across all fungal types represented. We believe this enhancement strengthens the manuscript by providing a more complete and accurate depiction of the strategies employed by these fungi during host colonization. Please see the revised figure in the updated manuscript.

Moreover, in Table 4 in tabular form, we presented the strategies employed by different BF, detailing their penetration mechanisms, key effectors, nutrient acquisition mechanisms, and reproduction strategies. This table enhances the understanding of how various fungi interact with host plants and adapt their infection strategies. Presenting this information in a structured and detailed format provides a clear and accessible reference for researchers studying plant-pathogen interactions. The comprehensive overview of effector-mediated suppression mechanisms and the specific examples of fungal strategies will aid in the development of new approaches to enhance plant resistance to fungal pathogens.

Comments 6: The information on lines 290-327 should also be presented in a table like Table 4.

Response 6: With the new table made in Comments 3 all information on lines 290-327 is condensed into Table 2.

Comments 7: Figure 2. In the caption to the figure, decipher the names of the proteins shown schematically in the figure.

Response 7: Thank you for your insightful comments regarding Figure 2. I appreciate your suggestion to clarify the names of the proteins depicted in the figure. I have revised the figure legend to include a detailed explanation of the abbreviations and symbols representing each protein. This should help readers better understand the molecular interactions illustrated in the figure. Please, read the LNs 455-461.

Figure 2. Effector proteins repertoire secreted by biotrophic fungi for suppressing plant defense responses. A. Effector secreted by rust fungi: AvrSr35, AvrSr43, AvrSr45, PgtSR1, Ps87, PEC6, PNPi, Pst02549, Pst18220, Pst03196, Pst05023, Shr1, PstSCR1, PSEC2, PstCFEM1, PstCEP1, Uaca family (1-46), Ua-RTP1, Ua-RTP1.

Effector secreted by powdery mildew species: AVRA1, Avrk1, Avr10, BEC1016, CSEP0064/BEC1054, CSEP0264/BEC1011, CSEP0105, CSEP087, EqCSEP01276.

Effector secreted by smut fungi: Hum3, Rsp1, Pit2, Pep1, Tin2, See1, Shy1, CMU1, ThSCSP_12, gsr1, uan2, g2666, g3970, g6610, g1513, g3890, g4549, g1052, g1084, g4554, g5159.

Comments 8: Figure 3. What do the symbols 1-4, A-D, W-Z mean? It is necessary to specify the names in the caption of the figure.

Response 8: Thank you for your valuable feedback on Figure 3. We understand the importance of clearly defining the symbols used in the figure for better comprehension. We have revised the figure caption to include a detailed explanation of what each symbol (1-4, A-D, W-Z) represents. Below is the updated caption:

Figure 3. Plant Receptor Proteins (R) involved in the perception of biotrophic pathogenic fungi to induce plant defense responses.

  1. Rust Fungi (blue color):
  • Symbols 1-4 represent different plant receptor proteins involved in rust disease resistance:
    • Symbol 1: LRp1 (Leucine-rich repeat protein 1)
    • Symbol 2: MLR46 (Multigene family member 46)
    • Symbol 3: NRfg1 (Nucleotide-binding site-leucine-rich repeat receptor 1)
    • Symbol 4: Rfg4 (Resistance gene 4)
  1. Powdery Mildew Fungi (green color):
  • Symbols A-D represent different plant receptor proteins involved in powdery mildew resistance:
    • Symbol A: Mla (Mildew resistance locus A)
    • Symbol B: Mlo (Mildew locus O)
    • Symbol C: RPW8.1 (Resistance to powdery mildew 8.1)
    • Symbol D: RPW8.2 (Resistance to powdery mildew 8.2)
  1. Smut Fungi (orange color):
  • Symbols W-Z represent different plant receptor proteins involved in smut disease resistance:
    • Symbol W: Ruh1 (Resistance to Ustilago hordei 1)
    • Symbol X: Ut11 (Ustilago tritici resistance 11)
    • Symbol Y: FRT-Hyg (Fungus resistance gene transformed with Hygromycin)
    • Symbol Z: FRT-Nat (Fungus resistance gene transformed with Nourseothricin)

This revised caption now clearly defines the symbols, giving readers an accurate understanding of the specific plant receptor proteins depicted in Figure 3. We believe these enhancements will improve the clarity and overall quality of the manuscript. Thank you for helping us make this improvement.

Comments 9: The sentences on lines 397-398 and 451-454 have the same semantic meaning, they should be rewritten or deleted.

Response 9: Please, read that LNs 451-454 will be deleted

Comments 10: The information on lines 473-493 should also be presented in a table similar to Table 4.

Response 10: Thank you for your insightful suggestion. In response to your recommendation, we have created Table 7 to provide a structured overview of how biotrophic fungi (BF) manipulate various plant hormones, such as Jasmonic Acid (JA), Salicylic Acid (SA), Abscisic Acid (ABA), and Ethylene (ETH), to suppress plant defense responses. This table outlines the mechanisms employed by the fungi, their effects on defense responses, and specific examples of hormone-fungus interactions. The table's structured format highlights the complex strategies BF employ to manipulate plant hormonal pathways, thus facilitating their colonization and pathogenesis.

Please, read the LNs 577-584.

Table 7 provides a structured overview of how BF manipulate various plant hormones, including JA, SA, ABA, and ETH, to suppress plant defense responses. It includes the mechanisms employed by the fungi, their effects on defense responses, and specific examples or details for each hormone-fungus interaction. This table highlights the intricate strategies BF employ to manipulate plant hormonal pathways, thereby facilitating their colonization and pathogenesis.

 Comments 11: The sentence on line 510 should be moved to the beginning of the chapter.

Response 11: Thank you for your valuable suggestion. We have revised the manuscript according to your recommendation. The sentence originally found on line 510 has been moved to the beginning of the chapter to enhance the logical flow and coherence of the content.

Comments 12: Perhaps another chapter should be added describing the evolutionary aspect of the relationship between biotrophic fungi and plants. How pathogens have learned to activate or reduce the plant's defense response.

Response 12: We appreciate the reviewer's insightful suggestion to include a Section on the evolutionary relationship between biotrophic fungi (BF) and plants. In response, we have incorporated a new Section, Section 7: Co-evolutionary Dynamic between BF and Plants, into the manuscript (read the LNs 733-910). This chapter provides a comprehensive overview of the co-evolutionary arms race between BF and plant defense mechanisms. It discusses how these ancient and intricate associations have evolved over millions of years, leading to the diversification of plant resistance traits and the development of sophisticated fungal strategies to overcome these defenses.

In this section, we delve into key plant defense mechanisms such as cell wall fortification, production of antimicrobial compounds, and activation of signaling pathways like salicylic acid (SA) and jasmonic acid (JA). Additionally, we explore the evolutionary adaptations of BF, including the diversification of their effector repertoires, stealthy colonization strategies, and mechanisms for nutrient acquisition.

The inclusion of this section enhances our manuscript by providing crucial context on the dynamic co-evolutionary processes that shape host-pathogen interactions, thereby contributing to a deeper understanding of how these relationships inform strategies for sustainable crop protection.

Reviewer 3 Report

The review contains data on interactions of biotrophic fungi (BF) with plants, in particular the ability of these plant pathogens to secrete effectors allowing biotrophs to suppress PAMP-triggered immunity, escape from the plant defense responses and thus facilitate infection. In addition, the capability of these fungi to induce ETI after BF effectors' recognition by plant receptors is described, and possibilities of BF to manipulate the plant signaling via diverse influence on plant hormones functioning as signal molecules is showed.

Although all aforementioned properties of BF are well-known and deeply considered in a range of special reviews, I think, the submitted paper might be useful for readers, who start studying the modes of molecular BF-plant interactions and are interested in general information about such investigations. The main review advantage is that authors analyze publications of recent years.

However, the current version of the review requires additional editing and corrections.

Firstly, I would suggest authors to try to better systemize data on different pathogens. The text is replete with repetitions.

Secondly, I would recommend considering mechanisms of BF impact on plants in more detail. It would be nice to specify some of them. Uncovering the mechanisms implies not only "what BF do" but also "how they do this". For instance, it would be desirable to go beyond stating that BF target key enzymes of the SA biosynthesis or key components of SA- or JA-dependent signaling system, key enzymes of the SA biosynthesis, etc., but also to indicate these enzymes and components, where they are known, or to mention putative ones, if they are not identified so far. At least, authors could guide readers to related previous reviews.

Some examples need to be clarified, e.g., Lr10 gene encoding CC-NBS-LRR type of protein confer the resistance to leaf rust (Puccinia triticana), not to stem rust (P. graminis). I did not find any data that the LRR-RLK encoded by Lr10 involves in P. graminis recognition (lines 138-139) in #38 indicated as the reference supporting these data.

Table 2 illustrating the induction of plant defense responses has the same title as Table1 that is related to their suppression. In my opinion, the figures request improvement (very small illegible legends), and their captions need additional explanations (e.g., figure 3: what receptors are indicated by numbers 1-4 and letters A-D, W-Z).

I would also suggest reduce in text amount of numerous repeats of final goals of BF (facilitation of colonization, the escape of activation of defense responses or their weakening, establishing the compatibility and providing with nutratients, etc.), which uniformly end many paragraphs containing examples of the action of certain pathogens.  These similarities for biotrophs are evident from the Introduction and may be additionally highlighted there.

Additionally, the style of a range of phrases should be improved to avoid misunderstanding, confusion and double meaning. See, please, some (but not all) examples in Minor Comments.

Minor comments and suggestions are listed below

Line 81/ Secreting Secreted molecules called effectors (small proteins and some non-proteinaceous molecules) can be injected into plant cells…

Line 84/ … mechanisms (Effector-triggered immunity, ETI). ETI is a specific branch of the plant immune….

Line 90/ … are proteins or small molecules… Please indicate what non-proteinaceous or small molecules (RNA? secondary metabolites?)

Line 150/ REN1 (Resistance to Erysiphe necator 1): REN1 is an LRR-RLK that confers resistance

Lines 153-154/ Also, PMR4 (Powdery Mildew Resistance 4): PMR4 is represents an LRR-RLK identified in Arabidopsis thaliana …

Lines 153-164 and 168/ Pm3 (Powdery mildew resistance 3): Pm3 genes in wheat (Triticum aestivum) encode LRR-RLKs that confer resistance against B. graminis f.sp. tritici, …. Mlo (Mildew resistance locus o): The Mlo gene in barlely…

Line 174/ Leucine-Rich Repeat Receptors – this full name should be shown above, when first mentioned.

Line 176/ …the wheat genes Sr33 and  Sr35, which confer resistance against specific rust pathogens… What does " specific rust pathogens" mean? Did you mean “specific resistance against races of stem rust? The carrent writing looks like Sr33 and  Sr35 cofer resistance to several(?!) rust fungi specific to wheat, not only to stem rust pathogen. Are you sure that it is right?

Lines 185-186/ May be:specific resistance against some specific races of barley stem rust pathogen (P. graminis f.sp. hordei)?

  Lines from 190-194 and lines 200-202/ It seems that data on RPW8-NLR and PMR4-NLR would be better to combine in common paragraph.

Line 224, 226/Regarding ”BF’s effectors…”  and “the plant’s gene expression…” ( “plant’s” is repeated in many sentences, e.g. line 268) What is the form with an apostrophe used for? That is unnecessary. If you want to emphasize the plural, (effectors of biotrophs, genes of plants), you should write BFs’, plants’ (the plants’ gene expression, plants’ ability, etc.). Check throughout the text and correct.

Line 252/ In The table 1 summarizing summarizes the key points and….. responses, This table provides structured…

Table 1./ Remove unnecessary capital letters in the table title and in the point 3

Line 267/…plant defense responses resulting in ETI or of ETI

Line 290/For instance, Sr33 and Sr35 are two R genes that provide….

Line 361/ “…these BF” Which are ‘’these’’?

Line 393 and throughout the text/ Please check usage of “can”, “may” and ‘’might”.

Line 424/ What does "signaling of SA" mean? Is it SA-dependent signal transduction?

Line 463/ How can suppression of SA production produce effectors?

Author Response

For review article:

Manuscript ID jof-3063704: Uncovering the Mechanisms: The Role of Biotrophic Fungi in Activating or Suppressing Plant Defense Responses

Response to Reviewer 3 Comments

  1. Summary: Thank you very much for taking the time to review this manuscript. Please find the detailed responses below and the corresponding revisions/corrections highlighted/in track changes in the re-submitted files.
  2. Point-by-point response to Comments and Suggestions for Authors: Major comments: The review contains data on interactions of biotrophic fungi (BF) with plants, in particular the ability of these plant pathogens to secrete effectors allowing biotrophs to suppress PAMP-triggered immunity, escape from the plant defense responses and thus facilitate infection. In addition, the capability of these fungi to induce ETI after BF effectors' recognition by plant receptors is described, and the possibilities of BF to manipulate the plant signaling via diverse influences on plant hormones functioning as signal molecules is shown. Although all aforementioned properties of BF are well-known and deeply considered in a range of special reviews, I think, the submitted paper might be useful for readers, who start studying the modes of molecular BF-plant interactions and are interested in general information about such investigations. The main review advantage is that authors analyze publications of recent years. However, the current version of the review requires additional editing and corrections.
  1. Response to major comments: The authors appreciate the reviewer's thorough feedback and recognize the importance of addressing the issues raised to enhance the quality and clarity of our manuscript. As noted by the reviewer, our manuscript's main advantage lies in the analysis of recent publications. We have made additional efforts to emphasize how recent studies have contributed to our understanding of BF-plant interactions. In particular, we have updated sections to better integrate recent advances, providing a clearer connection between foundational concepts and cutting-edge research.

Below, we provide specific responses to address the concerns raised:

Reply to Detailed comments

Comments 1. Authors should rewrite the beginning of the introduction because the meaning of the sentences is repeated. For example, the meaning of the sentence on line 34 repeats the meaning of the sentence on lines 38-39.

Firstly, I would suggest authors to try to better systemize data on different pathogens. The text is replete with repetitions.

A new version of the manuscript is provided. 

Response 1: Systemization of Data on Different Pathogens:

We acknowledge that the current structure may have led to some repetition and lack of clarity in distinguishing between pathogens. We have restructured the manuscript to systematize better the data on different biotrophic fungi (BF) species. Specifically, we have reorganized sections [1. Introduction; and 2.0 Mechanisms of BF Impact on Plant Hosts] to present information more cohesively, minimizing redundancy and ensuring a clearer distinction between the discussed pathogens. In the Introduction section (please, read LNs 35-67), we have organized the data into distinct subsections based on the major biotrophic fungal groups: rust fungi, powdery mildew fungi, and smut fungi. Each subsection now highlights key characteristics of the respective fungal group, their host range, and relevant pathogenic mechanisms. In the second section "Mechanisms of BF Impact on Plant Hosts" (please, read LNs 90-184). We have reorganized the section to systematize the descriptions of mechanisms across different BF groups. Rather than repeating general concepts (e.g., penetration, effector secretion, and antioxidant enzyme manipulation) for each fungal group, we introduced these mechanisms in a more generalized manner and then elaborated on specific examples where necessary. Each subsection now focuses on distinct mechanisms or structures (e.g., specialized structures, effector proteins, antioxidant enzymes), with examples from different BF species integrated where relevant. This approach prevents the repetition of overarching concepts while allowing us to highlight pathogen-specific details. The examples of effector molecules presented in Table 1 have been revised for clarity, ensuring that unique aspects of each fungal group are emphasized without redundancy. Additionally, we have cross-referenced these examples with the text to avoid unnecessary repetition of information already presented in the table.

These revisions improve the flow of information and enhance the systemization of data on different pathogens, providing a more organized and coherent discussion in both sections. We hope this restructuring addresses the reviewer's concerns effectively.

Comments 2: Secondly, I would recommend considering mechanisms of BF impact on plants in more detail. It would be nice to specify some of them. Uncovering the mechanisms implies not only "what BF do" but also "how they do this".

Response 2: Mechanism 1. Developing specialized structures:

Appressoria

In rust fungi urediniospores or teliospores develop germ tube, it senses physical and chemical cues from the host plant's surface (1). Germ tube tip swells and develops into a specialized structure called an appressorium that is typically a flattened, melanized (darkly pigmented) with and adher to the host plant's surface (2).

Within the appressorium, a specialized infection structure called a penetration peg (narrow, hypha-like structure that extends from the appressorium and penetrates the host plant's cell wall) (3) by a combination of mechanical (high internal turgor pressure) and enzymatic activity (cutinase, cellulase, pectinase, xylanase and protease) to breach the host plant's cell wall, facilitating the invasion of the host tissue (4).

Once the penetration peg has successfully entered the host plant's cell, it develops a specialized infection structure called a haustorium that allows to obtain nutrients from the host plant's cells (5) and secretion of effector proteins to the host cytoplasm enabling the fungus to establish a successful infection (6).

Powdery mildew fungus appressoria are generally more flattened and are heavily pigmented, often appearing dark brown or melanized with an adhesive ring to adhere tightly to the host plant's surface (7). Penetration peg forms within the appressorium and then breaches the cell wall, allowing the fungus to enter the host plant's cells (8). Durin infection of Blumeria graminis enzymes like glycoside hydrolase and degradation-associated carbohydrate-active enzymes were recorded (9).

Smut fungi when teliospore germinate for a promycelium and undergo mitotic divisions, eventually budding off haploid cells known as sporidia. When two compatibles haploid sporidia detect each other through pheromone signaling and develop conjugation tubes directed towards one another (10). Smut fungi then sense plant surface cues and induce specialized infection structures called appressoria, with transmembrane receptors Sho1 and Msb2 playing essential roles in perceiving external stimuli (11). To breach the plant cell wall, smut fungi secrete various plant cell wall-degrading enzymes like (endo-β-1,4-glucanases, exo-β-1,4-glucanases (12), xylanases (13), mannanases (14), β-glucosidases (15), polygalacturonases (16), pectin lyases (17), and pectate lyases (18).

References

  • Staples, R. C., & Hoch, H. C. (1997). Physical and chemical cues for spore germination and appressorium formation by fungal pathogens. In Plant Relationships: Part A(pp. 27-40). Berlin, Heidelberg: Springer Berlin Heidelberg.
  • Dean, R. A. (1997). Signal pathways and appressorium morphogenesis. Annual review of phytopathology35(1), 211-234.
  • KHARAYAT, B. S. (2023). DISEASES OF FIELD AND HORTICULTURAL CROPS AND THEIR MANAGEMENT VOLUME–II. PHI Learning Pvt. Ltd.
  • Chethana, K. T., Jayawardena, R. S., Chen, Y. J., Konta, S., Tibpromma, S., Abeywickrama, P. D., ... & Hyde, K. D. (2021). Diversity and function of appressoria. Pathogens10(6), 746.
  • Mapuranga, J., Zhang, N., Zhang, L., Chang, J., & Yang, W. (2022). Infection strategies and pathogenicity of biotrophic plant fungal pathogens. Frontiers in Microbiology13, 799396.
  • Ryder, L. S., Cruz-Mireles, N., Molinari, C., Eisermann, I., Eseola, A. B., & Talbot, N. J. (2022). The appressorium at a glance. Journal of Cell Science135(14), jcs259857.
  • Lucas, J. A., Dyer, P. S., & Murray, T. D. (2000). Pathogenicity, host-specificity, and population biology of Tapesia spp., causal agents of eyespot disease of cereals.
  • Howard, R. J. (1997). Breaching the outer barriers—cuticle and cell wall penetration. In Plant Relationships: Part A(pp. 43-60). Berlin, Heidelberg: Springer Berlin Heidelberg.
  • Pham, T. A., Kyriacou, B. A., Schwerdt, J. G., Shirley, N. J., Xing, X., Bulone, V., & Little, A. (2019). Composition and biosynthetic machinery of the Blumeria graminis f. sp. hordei conidia cell wall. The Cell Surface5, 100029.
  • Garcia-Pedrajas, M. D., Klostermann, S. J., Andrews, D. L., & Gold, S. E. (2004). The Ustilago maydis-maize interaction. Plant–Pathogen interactions11, 166-201.
  • Xia, W., Yu, X., & Ye, Z. (2020). Smut fungal strategies for the successful infection. Microbial pathogenesis142, 104039.
  • Moreno-Sánchez, I., Pejenaute-Ochoa, M. D., Navarrete, B., Barrales, R. R., & Ibeas, J. I. (2021). Ustilago maydis secreted endo-xylanases are involved in fungal filamentation and proliferation on and inside plants. Journal of fungi7(12), 1081.
  • Xia, W., Yu, X., & Ye, Z. (2020). Smut fungal strategies for the successful infection. Microbial pathogenesis142, 104039.
  • Kubicek, C. P., Starr, T. L., & Glass, N. L. (2014). Plant cell wall–degrading enzymes and their secretion in plant-pathogenic fungi. Annual review of phytopathology, 52(1), 427-451.
  • Geiser, E., Reindl, M., Blank, L. M., Feldbrügge, M., Wierckx, N., & Schipper, K. (2016). Activating intrinsic carbohydrate-active enzymes of the smut fungus Ustilago maydis for the degradation of plant cell wall components. Applied and environmental microbiology82(17), 5174-5185.
  • Castruita‐Domínguez, J. P., González‐Hernández, S. E., Polaina, J., Flores‐Villavicencio, L. L., Alvarez‐Vargas, A., Flores‐Martínez, A., ... & Leal‐Morales, C. A. (2014). Analysis of a polygalacturonase gene of Ustilago maydis and characterization of the encoded enzyme. Journal of Basic Microbiology54(5), 340-349.
  • Soria, N. W., Figueroa, A. C., Díaz, M. S., Alasino, R. V., Yang, P., & Beltramo, D. M. (2022). Identification and expression of some plant cell wall-degrading enzymes present in three ontogenetics stages of Thecaphora frezii, a peanut (Arachis hypogaea L.) pathogenic fungus.
  • Perlin, M. H., Amselem, J., Fontanillas, E., Toh, S. S., Chen, Z., Goldberg, J., ... & Cuomo, C. A. (2015). Sex and parasites: genomic and transcriptomic analysis of Microbotryum lychnidis-dioicae, the biotrophic and plant-castrating anther smut fungus. BMC genomics16, 1-24.

Mechanism 2. Secrete effector proteins in plant

Effector proteins that can be directed to the host chloroplast by mimicking the plant's own sorting signals and like suppressors of the host's RNA interference (RNAi) machinery, targeting conserved cellular proteins that are essential components of the plant's defense pathways (1). By disabling the host's RNAi system, the fungal effectors can effectively disrupt the plant's ability to organize a robust immune response, ultimately facilitating the pathogen's successful colonization and proliferation within the host (2).

Biotrophic fungal pathogens have evolved sophisticated effector proteins that can directly bind to the promoters of host defense genes, thereby modulating their transcriptional processes and leading to the suppression of the plant's immune responses (3). Furthermore, fungal effectors can also camouflage themselves as host modulators, diverting the metabolic flux of various compounds within the plant, resulting in a deficiency of crucial precursors or defense-related compounds (4).

BF induce the host to produce anti-cell death factors or hijack the plant's own cell death regulatory machinery to prevent the activation of programmed cell death, which would otherwise limit the pathogen's ability to establish a successful infection (5). By effectively suppressing cell death signaling, biotrophic fungi can create a hospitable environment within the host, allowing them to thrive and proliferate without triggering the plant's defensive cell death responses (6).

By suppressing ROS accumulation, biotrophic fungi can create a more favorable environment for their growth and proliferation within the host plant (7). Specific fungal effectors, such as TalSP and the nudix hydrolases TaNUDX23, have been identified as playing crucial roles in this process (8, 9).

Biotrophic fungal effectors can target and disrupt the host's pattern recognition receptors (PRRs) that are responsible for detecting pathogen-associated molecular patterns (PAMPs) (10). By interfering with the activation or signaling of these PRRs, the effectors can prevent the initiation of the PTI cascade, effectively shutting down the plant's first line of defense (11).

Additionally, some fungal effectors can directly inhibit the callose deposition process. Callose is a polysaccharide that is rapidly deposited at the site of attempted pathogen invasion, forming a physical barrier to restrict pathogen entry (12). Certain effectors may interfere with the enzymatic machinery responsible for callose synthesis or may trigger the host's own negative regulators of callose deposition (13).

Some effectors may promote the degradation of host's Adenosine kinase (ADKs) proteins, either by direct targeting or by inducing host processes that degrade the enzymes (14). The reduced levels of ADKs lead to a decrease in salicylic acid production and the subsequent suppression of PTI (15).

Fungal effectors can target and disrupt the host's machinery responsible for the formation of defense-related vesicles, such as the endoplasmic reticulum and Golgi apparatus (16). By interfering with the proper assembly and cargo loading of these vesicles, the effectors can prevent the transport and secretion of defense proteins (17), enzymes (18), and antimicrobial compounds (19).

BF effectors promote enhanced plasmodesmatal flux, facilitating the intercellular movement of nutrients and suppressing host defense responses (20). Modification in the structure or regulation of plasmodesmata, modified cytoplasmic channels that connect plant cells (21). By increasing the permeability and size exclusion limit of plasmodesmata, the fungal effectors can enable the passage of nutrients, signaling molecules, while limiting the transport of defense-related compounds (22).

Biotrophic fungi have evolved effector proteins that can directly interact with and manipulate host transcription factors to suppress plant defense responses (23). These effectors may bind to and sequester key transcription factors that are responsible for activating defense-related genes, preventing them from initiating the transcriptional programs necessary for the plant's immune response (24). Alternatively, the fungal effectors may act as transcriptional co-regulators, interfering with the ability of defense-related transcription factors to bind to and activate their target genes (25).

Fungal effectors may directly bind to and mask the Avr proteins, preventing their detection by the corresponding host resistance (R) proteins (26). Additionally, the effectors may interfere with the signaling pathways that normally lead to the activation of ETI upon Avr protein recognition, disrupting the downstream defense responses (27).

References

  • Sarmiento, C. (2008). Suppressors of RNA silencing in plants. Tallinn University of Technology.
  • Sharma, M., Fuertes, D., Perez-Gil, J., & Lois, L. M. (2021). SUMOylation in phytopathogen interactions: Balancing invasion and resistance. Frontiers in cell and developmental biology9, 703795.
  • Jaswal, R., Kiran, K., Rajarammohan, S., Dubey, H., Singh, P. K., Sharma, Y., ... & Sharma, T. R. (2020). Effector biology of biotrophic plant fungal pathogens: Current advances and future prospects. Microbiological research241, 126567.
  • Han, X., & Kahmann, R. (2019). Manipulation of phytohormone pathways by effectors of filamentous plant pathogens. Frontiers in Plant Science10, 822.
  • Cheng, Y., Yao, J., Zhang, Y., Li, S., & Kang, Z. (2016). Characterization of a Ran gene from Puccinia striiformis f. sp. tritici involved in fungal growth and anti-cell death. Scientific Reports6(1), 35248.
  • Dickman, M. B., & de Figueiredo, P. (2013). Death be not proud—cell death control in plant fungal interactions. PLoS Pathogens9(9), e1003542.
  • Segal, L. M., & Wilson, R. A. (2018). Reactive oxygen species metabolism and plant-fungal interactions. Fungal Genetics and Biology110, 1-9.
  • Wang, X., Zhai, T., Zhang, X., Tang, C., Zhuang, R., Zhao, H., ... & Wang, X. (2021). Two stripe rust effectors impair wheat resistance by suppressing import of host Fe–S protein into chloroplasts. Plant Physiology187(4), 2530-2543.
  • Rao, S., Cao, H., O’Hanna, F. J., Zhou, X., Lui, A., Wrightstone, E., ... & Li, L. (2024). Nudix hydrolase 23 post-translationally regulates carotenoid biosynthesis in plants. The Plant Cell36(5), 1868-1891.
  • Wang, X., Jiang, N., Liu, J., Liu, W., & Wang, G. L. (2014). The role of effectors and host immunity in plant–necrotrophic fungal interactions. Virulence5(7), 722-732.
  • Noman, A., Aqeel, M., & Lou, Y. (2019). PRRs and NB-LRRs: from signal perception to activation of plant innate immunity. International journal of molecular sciences20(8), 1882.
  • Wang, Y., Li, X., Fan, B., Zhu, C., & Chen, Z. (2021). Regulation and function of defense-related callose deposition in plants. International Journal of Molecular Sciences22(5), 2393.
  • Liu, J., Zhang, L., & Yan, D. (2021). Plasmodesmata-involved battle against pathogens and potential strategies for strengthening hosts. Frontiers in Plant Science12, 644870.
  • Jaswal, R., Kiran, K., Rajarammohan, S., Dubey, H., Singh, P. K., Sharma, Y., ... & Sharma, T. R. (2020). Effector biology of biotrophic plant fungal pathogens: Current advances and future prospects. Microbiological research241, 126567.
  • Liu, C., Pedersen, C., Schultz‐Larsen, T., Aguilar, G. B., Madriz‐Ordeñana, K., Hovmøller, M. S., & Thordal‐Christensen, H. (2016). The stripe rust fungal effector PEC 6 suppresses pattern‐triggered immunity in a host species‐independent manner and interacts with adenosine kinases. New Phytologist.
  • Tariqjaveed, M., Mateen, A., Wang, S., Qiu, S., Zheng, X., Zhang, J., ... & Sun, W. (2021). Versatile effectors of phytopathogenic fungi target host immunity. Journal of Integrative Plant Biology63(11), 1856-1873.
  • Abubakar, Y. S., Sadiq, I. Z., Aarti, A., Wang, Z., & Zheng, W. (2023). Interplay of transport vesicles during plant-fungal pathogen interaction. Stress biology3(1), 35.
  • Abubakar, Y. S., Sadiq, I. Z., Aarti, A., Wang, Z., & Zheng, W. (2023). Interplay of transport vesicles during plant-fungal pathogen interaction. Stress biology3(1), 35.
  • Brown, N. A., & Hammond‐Kosack, K. E. (2015). Secreted biomolecules in fungal plant pathogenesis. Fungal biomolecules: sources, applications and recent developments, 263-310.
  • Rahman, M. S., Madina, M. H., Plourde, M. B., Dos Santos, K. C. G., Huang, X., Zhang, Y., ... & Germain, H. (2021). The fungal effector Mlp37347 alters plasmodesmata fluxes and enhances susceptibility to pathogen. Microorganisms9(6), 1232.
  • Brown, N. A., & Hammond‐Kosack, K. E. (2015). Secreted biomolecules in fungal plant pathogenesis. Fungal biomolecules: sources, applications and recent developments, 263-310.
  • Tabassum, N., & Blilou, I. (2022). Cell-to-cell communication during plant-pathogen interaction. Molecular Plant-Microbe Interactions35(2), 98-108.
  • Wang, X., Jiang, N., Liu, J., Liu, W., & Wang, G. L. (2014). The role of effectors and host immunity in plant–necrotrophic fungal interactions. Virulence5(7), 722-732.
  • Mathan, L., Dubey, N., Verma, S., & Singh, K. (2022). Transcription factors associated with defense response against fungal necrotrophs. In Transcription Factors for Biotic Stress Tolerance in Plants(pp. 61-78). Cham: Springer International Publishing.
  • Kumari, P., Ojha, R., Varshney, V., Gupta, V., & Salvi, P. (2024). Transcription Factors and Their Regulatory Role in Plant Defence Response. In Biotechnological Advances for Disease Tolerance in Plants(pp. 337-362). Singapore: Springer Nature Singapore.
  • Kanja, C., & Hammond‐Kosack, K. E. (2020). Proteinaceous effector discovery and characterization in filamentous plant pathogens. Molecular Plant Pathology21(10), 1353-1376.
  • Chen, J., Zhang, X., Rathjen, J. P., & Dodds, P. N. (2022). Direct recognition of pathogen effectors by plant NLR immune receptors and downstream signalling. Essays in biochemistry66(5), 471-483.

Mechanism 3. Activation or deactivating antioxidants enzymes

Blumeria graminis fs.p. hordei accumulates 3-hydroxykynurenine, a redox-active substance, which facilitates the cross-linking of the pathogen to the host surface (1). Bgh also expresses a secreted catalase enzyme that is essential for removing hydrogen peroxide produced by the host plant (2). The removal of hydrogen peroxide by catalase prevents the host from cross-linking its cell wall as a defense mechanism against pathogen penetration (3).

Some rust fungi have evolved the ability to produce their own SOD enzymes. This allows them to neutralize the ROS produced by the plant, reducing the oxidative stress on the fungus (4). By scavenging superoxide radicals, the fungal SOD can help the pathogen evade or suppress the plant's initial defense response, allowing the infection to become established (5, 6).

Glycine-serine-rich effector, PstGSRE4, produced by the wheat stripe rust fungus Puccinia striiformis f. sp. tritici (Pst) has been shown to inhibit the enzymatic activity of the wheat copper-zinc superoxide dismutase (TaCZSOD2) (7). This wheat superoxide dismutase enzyme acts as a positive regulator, enhancing the plant's resistance against the Pst pathogen (8). By targeting and suppressing the activity of TaCZSOD2, PstGSRE4 effectively disrupts the host's ability to mount an effective oxidative defense response (7). This strategy enables the rust fungus to evade or dampen the plant's initial defense mechanisms, facilitating the establishment and progression of the infection.

Ustilago maydis, has evolved a highly effective virulence mechanism mediated by its secreted effector protein, Pep1. Studies have demonstrated that Pep1 plays a crucial role in suppressing the host's oxidative burst response during the early stages of infection (9). Mechanistically, Pep1 functions by directly inhibiting the activity of apoplastic peroxidases in the plant's extracellular space. Peroxidases are key enzymes involved in the generation of reactive oxygen species (ROS), which are typically produced by the host as a defense response against invading pathogens (10). By neutralizing the activity of these apoplastic peroxidases, the Pep1 effector effectively dampens the plant's ability to mount an oxidative burst, a critical component of the innate immune system (11). Some powdery mildew species have been found to secrete their own peroxidase enzymes, which can break down the H2O2 and other ROS produced by the plant (12). By neutralizing the ROS, the fungal peroxidases can help the pathogen evade or suppress the plant's initial oxidative defense response, allowing the infection to establish and progress (9).

Blumeria graminis f. sp. hordei secreted catalase enzyme that plays a crucial role in the removal of hydrogen peroxide (H2O2) produced by the plant as part of its oxidative defense (13). By effectively neutralizing the H2O2, the fungal catalase enable growth and spread during host infection (14).

References

  • Wilson, T. G., Thomsen, K. K., Petersen, B. O., Duus, J. Ø., & Oliver, R. P. (2003). Detection of 3-hydroxykynurenine in a plant pathogenic fungus. Biochemical Journal371(3), 783-788.
  • Yuan, H., Jin, C., Pei, H., Zhao, L., Li, X., Li, J., ... & Shen, Q. H. (2021). The powdery mildew effector CSEP0027 interacts with barley catalase to regulate host immunity. Frontiers in Plant Science12, 733237.
  • Hückelhoven, R. (2007). Cell wall–associated mechanisms of disease resistance and susceptibility.  Rev. Phytopathol.45(1), 101-127.
  • Zheng, P., Chen, L., Zhong, S., Wei, X., Zhao, Q., Pan, Q., ... & Liu, J. (2020). A Cu‐only superoxide dismutase from stripe rust fungi functions as a virulence factor deployed for counter defense against host‐derived oxidative stress. Environmental Microbiology22(12), 5309-5326.
  • Liu, X., & Zhang, Z. (2022). A double‐edged sword: reactive oxygen species (ROS) during the rice blast fungus and host interaction. The FEBS journal289(18), 5505-5515.
  • Zheng, P., Chen, L., Zhong, S., Wei, X., Zhao, Q., Pan, Q., ... & Liu, J. (2020). A Cu‐only superoxide dismutase from stripe rust fungi functions as a virulence factor deployed for counter defense against host‐derived oxidative stress. Environmental Microbiology22(12), 5309-5326.
  • Liu, C., Wang, Y., Wang, Y., Du, Y., Song, C., Song, P., ... & Guo, J. (2022). Glycine-serine-rich effector PstGSRE4 in Puccinia striiformis f. sp. tritici inhibits the activity of copper zinc superoxide dismutase to modulate immunity in wheat. PLoS Pathogens18(7), e1010702.
  • Feng, H., Liu, W., Zhang, Q., Wang, X., Wang, X., Duan, X., ... & Kang, Z. (2014). TaMDHAR4, a monodehydroascorbate reductase gene participates in the interactions between wheat and Puccinia striiformis f. sp. tritici. Plant physiology and biochemistry76, 7-16.
  • Hemetsberger, C., Herrberger, C., Zechmann, B., Hillmer, M., & Doehlemann, G. (2012). The Ustilago maydis effector Pep1 suppresses plant immunity by inhibition of host peroxidase activity. PLoS pathogens8(5), e1002684.
  • Doehlemann, G., Van Der Linde, K., Aßmann, D., Schwammbach, D., Hof, A., Mohanty, A., ... & Kahmann, R. (2009). Pep1, a secreted effector protein of Ustilago maydis, is required for successful invasion of plant cells. PLoS pathogens5(2), e1000290.
  • Wang, Y., Pruitt, R. N., Nuernberger, T., & Wang, Y. (2022). Evasion of plant immunity by microbial pathogens. Nature Reviews Microbiology20(8), 449-464.
  • Yuan, H., Jin, C., Pei, H., Zhao, L., Li, X., Li, J., ... & Shen, Q. H. (2021). The powdery mildew effector CSEP0027 interacts with barley catalase to regulate host immunity. Frontiers in Plant Science12, 733237.
  • Yuan, H., Jin, C., Pei, H., Zhao, L., Li, X., Li, J., ... & Shen, Q. H. (2021). The powdery mildew effector CSEP0027 interacts with barley catalase to regulate host immunity. Frontiers in Plant Science12, 733237.
  • Zhang, Z., Henderson, C., & Gurr, S. J. (2004). Blumeria graminis secretes an extracellular catalase during infection of barley: potential role in suppression of host defence. Molecular plant pathology5(6), 537-547.

Comments 3: Some examples need to be clarified, e.g., Lr10 gene encoding CC-NBS-LRR type of protein confer the resistance to leaf rust (Puccinia triticana), not to stem rust (P. graminis). I did not find any data that the LRR-RLK encoded by Lr10 involves in P. graminis recognition (lines 138-139) in #38 indicated as the reference supporting these data.

Response 3: It is true that Lr10 recognize Puccinia triticana instead of Puccinia graminis. Sentence lines 138-139 will be deleted together with the respective reference.

Comments 4: Table 2 illustrating the induction of plant defense responses has the same title as Table1 that is related to their suppression. In my opinion, the figures request improvement (very small illegible legends), and their captions need additional explanations (e.g., figure 3: what receptors are indicated by numbers 1-4 and letters A-D, W-Z).

Response 4: Table 2 and figures were modified in the new version of manuscript

Comments 5: I would also suggest reduce in text amount of numerous repeats of final goals of BF (facilitation of colonization, the escape of activation of defense responses or their weakening, establishing the compatibility and providing with nutrients, etc.), which uniformly end many paragraphs containing examples of the action of certain pathogens.  These similarities for biotrophs are evident from the Introduction and may be additionally highlighted there.

Response 5: Modification in the entire manuscript have been done and in my opinion repeats have reduced.

Detail comments

Minor comments and suggestions are listed below

Comments 6: Line 81/ Secreting Secreted molecules called effectors (small proteins and some non-proteinaceous molecules) can be injected into plant cells…

Response 6: I agree with the suggestion.

Comments 7: Line 84/ … mechanisms (Effector-triggered immunity, ETI). ETI is a specific branch of the plant immune….

Response 7: I agree with the suggestion.

Comments 8: Line 90/ … are proteins or small molecules… Please indicate what non-proteinaceous or small molecules (RNA? secondary metabolites?)

Response 8: Small molecules like 3-hydroxykynurenine are produced by Blumeria graminis hordei facilitate the fungus' adhesion to the host surface or interfere with the plant's oxidative defenses (1). Secretion of small RNAs that target a wide range of host transcripts (2), including not only endogenous plant genes like those encoding transposons and kinases (3), but also transcripts from other kingdoms, such as those coding for crucial plant immune receptors like nucleotide-binding domain leucine-rich repeat (NLR) proteins (4), as well as various families of transcription factors involved in coordinating defense responses (5).

Secretion of microRNAs (miR444b.2, miR164a, miR319b, miR169, Osa-miR439, and miR396) that can target, and silence host plant genes were reported in Magnaporthe oryzae negatively regulate rice resistance to M. oryzae (6). Botrytis cinerea, the causal agent of grey mold, produces small regulatory RNAs (Bc-sRNAs) that can exploit the host plant's own RNA interference (RNAi) pathways to silence genes involved in immunity (7). These fungal-derived sRNAs achieve this by binding to the key Argonaute 1 (AGO1) protein in Arabidopsis and tomato, leading to the selective downregulation of host defense-related transcripts (8).

Magnaporthe oryzae, produces an active enzyme known as Ace1 that is likely involved in the biosynthesis of a secondary metabolite (9). It is well-established that fungal polyketide synthases (PKS) and non-ribosomal peptide synthetases (NRPS) are often responsible for the production of mycotoxins and host-specific toxins (HSTs) (10). The Ace1 enzyme produced by M. oryzae may be part of a biosynthetic pathway leading to the production of a secondary metabolite, which could potentially function as a host-specific toxin (11). By targeting and interfering with the host's defense mechanisms, such a toxin could enhance the virulence of the rice blast fungus and contribute to its successful colonization of the plant.

Fungal secondary metabolites classified as host-specific toxins (HSTs) represent prime examples of pathogen-derived effectors that exhibit selective toxicity towards their cognate plant hosts (12). Exemplary HSTs include victorin, HC toxin, depudecin, T-toxin, AAL toxin, destruxin, and ACR toxin (9). These secondary metabolites produced by various fungal species have been extensively studied for their role in modulating host plant defenses and facilitating the progression of pathogenic infections (13).

References

  • Hückelhoven, R. (2005). Powdery mildew susceptibility and biotrophic infection strategies. FEMS Microbiology letters245(1), 9-17.
  • Katiyar-Agarwal, S., & Jin, H. (2010). Role of small RNAs in host-microbe interactions. Annual review of phytopathology48(1), 225-246.
  • Huang, C. Y., Wang, H., Hu, P., Hamby, R., & Jin, H. (2019). Small RNAs–big players in plant-microbe interactions. Cell host & microbe26(2), 173-182.
  • Qiao, Y., Xia, R., Zhai, J., Hou, Y., Feng, L., Zhai, Y., & Ma, W. (2021). Small RNAs in plant immunity and virulence of filamentous pathogens. Annual Review of Phytopathology59(1), 265-288.
  • Fei, Q., Zhang, Y., Xia, R., & Meyers, B. C. (2016). Small RNAs add zing to the zig-zag-zig model of plant defenses. Molecular Plant-Microbe Interactions29(3), 165-169.
  • Gao, S., Hou, Y., Huang, Q., Wu, P., Han, Z., Wei, D., ... & Wang, J. (2023). Osa-miR11117 targets OsPAO4 to regulate rice immunity against the blast fungus Magnaporthe oryzae. International Journal of Molecular Sciences24(22), 16052.
  • Bitencourt, T. A., Pessoni, A. M., Oliveira, B. T., Alves, L. R., & Almeida, F. (2022). The RNA content of fungal extracellular vesicles: At the “cutting-edge” of pathophysiology regulation. Cells11(14), 2184.
  • Chen, A., Halilovic, L., Shay, J. H., Koch, A., Mitter, N., & Jin, H. (2023). Improving RNA-based crop protection through nanotechnology and insights from cross-kingdom RNA trafficking. Current opinion in plant biology, 102441.
  • Collemare, J., Pianfetti, M., Houlle, A. E., Morin, D., Camborde, L., Gagey, M. J., ... & Böhnert, H. U. (2008). Magnaporthe grisea avirulence gene ACE1 belongs to an infection‐specific gene cluster involved in secondary metabolism. New Phytologist179(1), 196-208.
  • Collemare, J., & Lebrun, M. H. (2011). Fungal secondary metabolites: ancient toxins and novel effectors in plant–microbe interactions. Effectors in plant–microbe interactions, 377-400.
  • Motoyama, T. (2020). Secondary metabolites of the rice blast fungus Pyricularia oryzae: Biosynthesis and biological function. International Journal of Molecular Sciences21(22), 8698.
  • Pradhan, A., Ghosh, S., Sahoo, D., & Jha, G. (2021). Fungal effectors, the double edge sword of phytopathogens. Current genetics67, 27-40.
  • Pusztahelyi, T., Holb, I. J., & Pócsi, I. (2015). Secondary metabolites in fungus-plant interactions. Frontiers in plant science6, 573.

Comments 9: Line 150/ REN1 (Resistance to Erysiphe necator 1): REN1 is an LRR-RLK that confers resistance

Response 9: I agree with the suggestion.

Comments 10: Lines 153-154/ Also, PMR4 (Powdery Mildew Resistance 4): PMR4 is represents an LRR-RLK identified in Arabidopsis thaliana …

Response 10: I agree with the suggestion.

Comments 11: Lines 153-164 and 168/ Pm3 (Powdery mildew resistance 3): Pm3 genes in wheat (Triticum aestivum) encode LRR-RLKs that confer resistance against B. graminis f.sp. tritici, …. Mlo (Mildew resistance locus o): The Mlo gene in barlely…

Response 11: I agree with the suggestion.

Comments 12: Line 174/ Leucine-Rich Repeat Receptors – this full name should be shown above, when first mentioned.

Response 12: I agree with the suggestion.

Comments 13: Line 176/ …the wheat genes Sr33 and  Sr35, which confer resistance against specific rust pathogens…

Response 13: What does " specific rust pathogens" mean? Did you mean “specific resistance against races of stem rust? The carrent writing looks like Sr33 and  Sr35 cofer resistance to several(?!) rust fungi specific to wheat, not only to stem rust pathogen. Are you sure that it is right?

The wheat resistance genes Sr33 and Sr35 confer race-specific resistance against Puccinia graminis f. sp. tritici where a specific resistance (R) gene, such as Sr33 or Sr35, recognizes a corresponding avirulence (Avr) effector molecule produced by a specific race or strain of the pathogen (1). This recognition triggers a robust defense response that effectively blocks the development and spread of the recognized pathogen race within the host plant. In the case of Sr33 (2) and Sr35 (3), these R genes have been shown to specifically recognize and respond to certain Avr effectors produced by various Puccinia graminis f. sp. tritici races (4). The resistance conferred by these genes is effective against some, but not all, stem rust pathogen races, as the pathogen can evolve to evade recognition by these R genes through mutations in the corresponding Avr genes (5).

References

  • Ellis, J. G., Lagudah, E. S., Spielmeyer, W., & Dodds, P. N. (2014). The past, present and future of breeding rust resistant wheat. Frontiers in plant science5, 641.
  • Lubega, J., Figueroa, M., Dodds, P. N., & Kanyuka, K. (2024). Comparative analysis of the avirulence effectors produced by the fungal stem rust pathogen of wheat. Molecular Plant-Microbe Interactions37(3), 171-178.
  • Lubega, J., Figueroa, M., Dodds, P. N., & Kanyuka, K. (2024). Comparative analysis of the avirulence effectors produced by the fungal stem rust pathogen of wheat. Molecular Plant-Microbe Interactions37(3), 171-178.
  • Sánchez-Martín, J., & Keller, B. (2021). NLR immune receptors and diverse types of non-NLR proteins control race-specific resistance in Triticeae. Current Opinion in Plant Biology62, 102053.
  • Hatta, M. A. M., Arora, S., Ghosh, S., Matny, O., Smedley, M. A., Yu, G., ... & Wulff, B. B. (2021). The wheat Sr22, Sr33, Sr35 and Sr45 genes confer resistance against stem rust in barley. Plant biotechnology journal19(2), 273-284.

Comments 14: Lines 185-186/ May be: …specific resistance against some specific races of barley stem rust pathogen (P. graminis f.sp. hordei)?

Response 14: Barley, like wheat, is susceptible to infection by the fungal pathogen Puccinia graminis f. sp. hordei, the causal agent of barley stem rust (1). Plants have evolved a variety of resistance (R) genes that can recognize and respond to specific pathogen effectors or molecules, triggering a defense response that prevents or limits disease development (2).

The resistance conferred by these R genes is often race-specific, meaning that the R gene can only recognize and respond to certain races or strains of the pathogen that carry the corresponding avirulence (Avr) gene or molecule (3). In the case of barley stem rust, specific R genes have been identified that can provide resistance against certain races of P. graminis f. sp. hordei, but not others (4). This is due to the evolutionary arms race between the plant and the pathogen, where the pathogen constantly evolves new virulence factors (Avr genes) to evade recognition by the host's R genes, while the plant acquires new R genes to maintain resistance (5).

For example, the barley resistance gene Rpg1 confers resistance against some, but not all, races of P. graminis f. sp. hordei (6). The Rpg1 protein recognizes and responds to specific Avr effectors produced by certain pathogen races, triggering a defense response that effectively blocks the development of the recognized races. However, as the pathogen evolves and acquires mutations in the Avr genes, it can become virulent against the Rpg1-mediated resistance, rendering the host plant susceptible to particular pathogen races (7).

References

  • Huerta-Espino, J., Singh, R. P., Roelfs, A. P., Misra, J. K., Tewari, J. P., Deshmukh, S. K., & Vágvölgyi, C. (2014). Rusts fungi of wheat. Fungi from different substrates27, 217-259.
  • Andersen, E. J., Ali, S., Byamukama, E., Yen, Y., & Nepal, M. P. (2018). Disease resistance mechanisms in plants. Genes9(7), 339.
  • Hammond-Kosack, K. E., & Kanyuka, K. (2007). Resistance genes (R genes) in plants. Encyclopedia of Life Science, 1-21.
  • Kleinhofs, A., Brueggeman, R., Nirmala, J., Zhang, L., Mirlohi, A., Druka, A., ... & Steffenson, B. J. (2009). Barley stem rust resistance genes: structure and function. The plant genome2(2).
  • Khavkin, E. E. (2021). Plant–pathogen molecular dialogue: Evolution, mechanisms and agricultural implementation. Russian Journal of Plant Physiology68, 197-211.
  • Hernandez, J., Del Blanco, A., Filichkin, T., Fisk, S., Gallagher, L., Helgerson, L., ... & Hayes, P. (2020). A Genome-Wide Association Study of Resistance to Puccinia striiformis f. sp. hordei and P. graminis f. sp. tritici in Barley and Development of Resistant Germplasm. Phytopathology110(5), 1082-1092.
  • Nirmala, J., Dahl, S., Steffenson, B. J., Kannangara, C. G., Von Wettstein, D., Chen, X., & Kleinhofs, A. (2007). Proteolysis of the barley receptor-like protein kinase RPG1 by a proteasome pathway is correlated with Rpg1-mediated stem rust resistance. Proceedings of the National Academy of Sciences104(24), 10276-10281.

Comments 15: Lines from 190-194 and lines 200-202/ It seems that data on RPW8-NLR and PMR4-NLR would be better to combine in common paragraph.

Response 15: New paragraph as a result of the combination of text placed from Lines 190-194 and lines 200-202.

The barley resistance gene Rpg1 encodes an NLR (nucleotide-binding, leucine-rich repeat) protein that confers race-specific resistance against certain races of the barley stem rust pathogen, Puccinia graminis f. sp. hordei. Rpg1 recognizes the specific AvrRpg1 effector produced by the rust fungus, triggering a robust defense response that effectively blocks the development of the recognized pathogen races. Similarly, the soybean resistance gene Rpp1 provides resistance against the Asian soybean rust pathogen, Phakopsora pachyrhizi, by recognizing the AvrRpp1 effector and activating defense mechanisms.

Comments 16: Line 224, 226/Regarding ”BF’s effectors…”  and “the plant’s gene expression…” ( “plant’s” is repeated in many sentences, e.g. line 268) What is the form with an apostrophe used for? That is unnecessary. If you want to emphasize the plural, (effectors of biotrophs, genes of plants), you should write BFs’, plants’ (the plants’ gene expression, plants’ ability, etc.). Check throughout the text and correct.

Response 16: I agree with the suggestion

Comments 17: Line 252/ In The table 1 summarizing summarizes the key points and….. responses, This table provides structured

Response 17: I agree with the suggestion

Comments 18: Table 1./ Remove unnecessary capital letters in the table title and in the point 3

Response 18: I agree with the suggestion

Comments 19: Line 267/…plant defense responses resulting in ETI or of ETI

Response 19: In my opinion "Plant defense responses resulting in ETI" is a more accurate and precise phrasing because it correctly conveys that the plant defense responses, triggered by the R protein-effector recognition, ultimately culminate in or lead to the activation of ETI.

Comments 20: Line 290/ For instance, Sr33 and Sr35 are two R genes that provide….

Response 20: The wheat resistance genes Sr33 and Sr35 confer resistance against the stem rust pathogen Puccinia graminis f. sp. tritici (Pgt)

Comments 21: Line 361/ “…these BF” Which are ‘’these’’?

Response 21: Rust, powdery mildew and smut fungi are ‘’these’’.

Comments 22: Line 393 and throughout the text/ Please check usage of “can”, “may” and ‘’might”.

Response 22: It will be checked.

Comments 23: Line 424/ What does "signaling of SA" mean? Is it SA-dependent signal transduction?

Response 24: "signaling of SA" refers to the signaling pathway involving the plant hormone salicylic acid (SA).

Comments 24: Line 463/ How can suppression of SA production produce effectors?

 Response 24: Puccinia effectors may target and disrupt components involved in SA perception or downstream signaling, such as SA receptors or transcription factors. By inhibiting SA signaling, the pathogen prevents the activation of defense-related genes and hampers the plants’ ability to mount an effective defense response against the biotrophic infection [134].

The suppression of salicylic acid (SA) production in plants cannot directly produce pathogen effectors. This is because effectors are molecules produced by the pathogen, not by the plant.

Round 2

Reviewer 2 Report

Authors significantly improved the manuscript. All my recommendations have been considered. However, before this manuscript can be published, some minor improvements should be done.

- Minor Essential Revisions:

1. The authors should improve the quality and size of Figure 1, which is now unreadable.

2. Also improve the quality of Figures 2 and 3.

3. Line 88,174: Remove "Biotrophic fungi", leave only "BF".

4. Line 197: Italicize Puccinia striiformis.

5. Line 214: H2O2 replace with H2O2.

6. Line 233: remove «reactive oxygen species, such as hydrogen peroxide», leave only the reduction «ROS», «H2O2».

7. Line 475-476: Italicize Ustilago hordei and Ustilago tritici.